# Deep learning linking mechanistic models to single-cell transcriptomics data reveals transcriptional bursting in response to DNA damage

Zhiwei Huang[1,2†], Songhao Luo[3†], Zihao Wang[1,2], Zhenquan Zhang[4], Benyuan Jiang[5]*, Qing Nie[3]*, Jiajun Zhang[1,2]*

[1]Guangdong Province Key Laboratory of Computational Science, Sun Yat-sen University, Guangzhou, China; [2]School of Mathematics, Sun Yat-sen University, Guangzhou, China; [3]Department of Mathematics, University of California Irvine, Irvine, United States; [4]School of Mathematics and Statistics, Guangdong University of Technology, Guangzhou, China; [5]Guangdong Lung Cancer Institute, Guangdong Provincial People's Hospital and Guangdong Academy of Medical Sciences, Guangzhou, China

*For correspondence:
jiangbenyuan@gdph.org.cn (BJ);
qnie@uci.edu (QN);
zhjiajun@mail.sysu.edu.cn (JZ)

†These authors contributed equally to this work

Competing interest: The authors declare that no competing interests exist.

## eLife Assessment

This study presents DeepTX, a **valuable** methodological tool that integrates mechanistic stochastic models with single-cell RNA sequencing data to infer transcriptional burst kinetics at genome scale. The approach is broadly applicable and of interest to subfields such as systems biology, bioinformatics, and gene regulation. The evidence supporting the findings is **solid**, with appropriate validation on synthetic data and thoughtful discussion of limitations related to identifiability and model assumptions.

**Abstract** Cells must adopt flexible regulatory strategies to make decisions regarding their fate, including differentiation, apoptosis, or survival in the face of various external stimuli. One key cellular strategy that enables these functions is stochastic gene expression programs. However, understanding how transcriptional bursting, and consequently, cell fate, responds to DNA damage on a genome-wide scale poses a challenge. In this study, we propose an interpretable and scalable inference framework, DeepTX, that leverages deep learning methods to connect mechanistic models and single-cell RNA sequencing (scRNA-seq) data, thereby revealing genome-wide transcriptional burst kinetics. This framework enables rapid and accurate solutions to transcription models and the inference of transcriptional burst kinetics from scRNA-seq data. Applying this framework to several scRNA-seq datasets of DNA-damaging drug treatments, we observed that fluctuations in transcriptional bursting induced by different drugs were associated with distinct fate decisions: 5'-iodo-2'-deoxyuridine treatment was associated with differentiation in mouse embryonic stem cells by increasing the burst size of gene expression, while low- and high-dose 5-fluorouracil treatments in human colon cancer cells were associated with changes in burst frequency that corresponded to apoptosis- and survival-related fate, respectively. Together, these results show that DeepTX enables genome-wide inference of transcriptional bursting from single-cell transcriptomics data and can generate hypotheses about how bursting dynamics relate to cell fate decisions.

## Introduction

Cells must employ flexible regulatory strategies to determine their fate, including cell differentiation, survival, or apoptosis, in response to diverse external stimuli (*López-Maury et al., 2008*; *Stadhouders et al., 2019*). One key cellular strategy that enables both beneficial and detrimental functions is the stochastic nature of gene expression programs, arising from intricate biochemical processes and termed 'noise' (*Raj and Oudenaarden, 2008*; *Balázsi et al., 2011*; *Losick and Desplan, 2008*). Noise can arise at many steps—from stochastic transcription factor binding and chromatin remodeling to variability in RNA transcription and protein translation. Among these diverse sources, transcriptional noise sits at the very start of the central dogma, converting upstream molecular fluctuations into the mRNA outputs that drive all downstream regulatory and functional processes. Transcriptional noise originates from a discontinuous pattern, wherein mRNA production occurs through alternating active and inactive gene states (*Eling et al., 2019*; *Tunnacliffe and Chubb, 2020*). This phenomenon, known as 'transcriptional bursting', has been observed in numerous experiments across both prokaryotic and eukaryotic cells (*Golding et al., 2005*; *Chubb et al., 2006*; *Larson et al., 2011*). The burst kinetics, characterized by burst frequency (BF) and burst size (BS), describe the stochastic behavior of molecular interactions at the gene expression process, offering valuable insights into how cells encode and leverage variability. However, it remains an unresolved question how cells make fate decisions in response to external signals through genome-wide regulation of transcriptional burst mechanisms (*Lammers et al., 2020*; *Rodriguez and Larson, 2020*).

DNA damage is a crucial and prevalent factor contributing to cellular responses, inducing varying degrees of cytotoxicity and thereby influencing diverse cell fate decisions (*Krenning et al., 2019*; *Su, 2006*; *Hafner et al., 2019*). For instance, low cytotoxicity limits cell proliferation (*van den Berg et al., 2019*; *Arora et al., 2017*; *Barr et al., 2017*; *Deng et al., 2023*), whereas heightened cytotoxicity

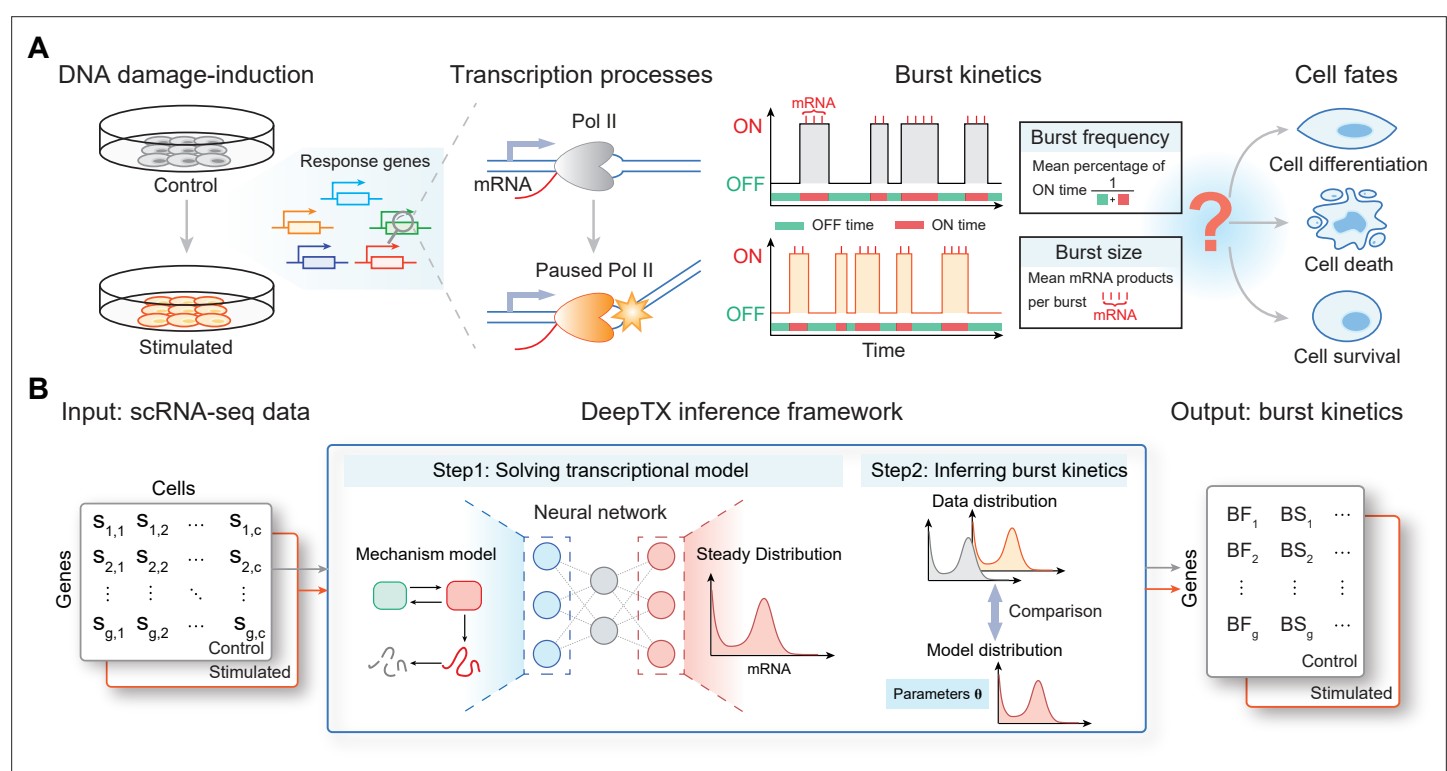

**Figure 1.** Overview of the DeepTX framework. (**A**) DNA-damaging drugs induce DNA double-strand damage, which will slow down or even stop the movement of RNA Pol II on the DNA double strands. Changes in the state of RNA Pol II movement will lead to changes in the kinetics of gene expression bursts (including burst frequency and burst size), which will affect cell fate decisions such as apoptosis, differentiation, and survival. (**B**) The input to the DeepTX framework is scRNA-seq data, and the output is the burst kinetics corresponding to each gene. The core of the DeepTX inference framework is a hierarchical model, which is a mixture of the stationary distribution of the mechanism model solved by the neural network (Step 1) and the binomial distribution followed by the sequencing process. This hierarchical model is used to infer the underlying dynamics parameters corresponding to the data distribution (Step 2).

prompts cell differentiation and senescence (*Müllers et al., 2014*; *Feringa et al., 2018*; *Toledo et al., 2008*; *Zhao et al., 2023*). Furthermore, extreme scenarios of high cytotoxicity can cause cells to engage in the apoptotic process (*Yousefzadeh et al., 2021*; *Carneiro and El-Deiry, 2020*; *Roos and Kaina, 2013*; *Zheng et al., 2018*). Experiments have shown that DNA damage can disrupt the gene transcription process (*Lans et al., 2019*). Specifically, DNA damage slows down the movement of RNA Pol II along the DNA strand (*Muñoz et al., 2009*). When more severe DNA damage is encountered, RNA Pol II exhibits a sliding pause behavior for DNA repair before resuming transcription (*Figure 1A*; *Gregersen and Svejstrup, 2018*; *Geijer and Marteijn, 2018*; *Giono et al., 2016*). These changes in RNA Pol II movement behavior would further reflect the cellular regulation of gene expression, particularly burst kinetics (*Friedrich et al., 2019*; *Calia et al., 2023*). A recent study has shown that cellular responses to DNA damage involve the regulation of BF on specific genes, rather than a genome-wide conclusion (*Friedrich et al., 2019*). They have demonstrated that cells respond to DNA damage by increasing gene expression noise, but they do not delve into the topic of transcriptional bursts (*Calia et al., 2023*; *Desai et al., 2021*). Therefore, the question of how DNA damage causes cells to regulate transcriptional burst kinetics on a genome-wide scale remains unresolved.

To understand the dynamics of transcriptional bursts, ideally, one needs to observe the fluctuations of gene expression over a continuous time interval, such as smFISH, scNT-seq, and HT-smFISH (*Femino et al., 1998*; *Qiu et al., 2020*; *Safieddine et al., 2023*). However, most existing techniques can only observe a limited number of genes, making their application to genome-wide studies challenging. Fortunately, single-cell RNA sequencing (scRNA-seq) technology provides excellent opportunities to explore this question, emerging as the leading method for genome-wide mRNA measurements and revealing gene expression noise within individual cells (*Tanay and Regev, 2017*; *Picelli et al., 2013*; *Zheng et al., 2017*). Many studies have employed scRNA-seq to elucidate the diverse sources of gene expression noise and the fundamental principles of transcriptional dynamics on a genome-wide scale (*Eling et al., 2019*; *Faure et al., 2017*; *Morgan and Marioni, 2018*; *Ochiai et al., 2020*). However, scRNA-seq data provide only static snapshots of cellular states. To infer the underlying transcriptional burst kinetics, one can apply mathematical models of stochastic gene expression, typically under the assumption that the observed distributions of scRNA-seq data reflect the steady-state outcome of the underlying dynamic processes (*Larsson et al., 2019*; *Luo et al., 2023a*; *Luo et al., 2023b*).

For mathematical modeling, the gene expression model for DNA damage ought to encompass more sophisticated generalized properties compared to the traditional telegraph model, which is a common model describing gene expression burst kinetics by genes switching stochastically between active (ON) and inactive (OFF) states with a single-step process (exponentially distributed waiting times) (*Stumpf et al., 2017*; *Voss and Hager, 2014*). However, the presence of DNA damage necessitates modeling the transcriptional process as a multi-step process, rather than a single-step process, to capture the additional complexity introduced by the damage (*Singh et al., 2013*; *Cavallaro et al., 2021*). Many efforts in this field are addressing the challenges of describing this multi-step process of gene expression in an interpretable and tractable model. One common modeling approach is to introduce multiple intermediate states between inactivate and activate states (*Zhang et al., 2012*; *Zhang and Zhou, 2014*; *Zhou and Zhang, 2012*). Although this multi-state model can fit the experimental data better, it is hard to rationalize the state numbers and parameters (*Desai et al., 2021*; *Rodriguez et al., 2019*; *Zoller et al., 2015*). Another alternative modeling approach is direct non-Markovian modeling for non-exponential waiting times in gene states, which maps the multiple parameters from multi-state models to a small number of interpretable parameters that are easily observed experimentally (*Soltani et al., 2015*; *Kumar et al., 2015*; *Schwabe et al., 2012*; *Stinchcombe et al., 2012*; *Zhang and Zhou, 2019a*; *Zhang and Zhou, 2019b*). However, the ensuing difficulty is solving the analytic solution of the steady distribution from a non-Markovian model, which may require some numerical simulation methods that are computationally resource-intensive and time-consuming, such as Monte Carlo methods (*Sisson et al., 2007*).

Statistical inference, particularly in recovering burst kinetic parameters from genome-wide scRNA-seq data, necessitates efficient and scalable inference algorithms (*Gómez-Schiavon et al., 2017*). Therefore, the stochastic dynamical system must be quickly solved as parameters are continuously updated throughout the inference process. Deep learning exhibits a broad array of applications as a contemporary method for addressing complex systems (*Jiang et al., 2021*; *Wang et al., 2019*; *Michoski et al., 2020*). For example, some studies employed neural networks to establish the

mapping relationship between a wide range of parameters in the gene expression model and the corresponding pre-simulated steady distribution (*Figure 2B*; *Jiang et al., 2021*; *Wang et al., 2019*; *Davis et al., 2020*; *Tang et al., 2023b*). This approach, based on neural networks, has been utilized for parameter inference in both deterministic and stochastic models (*Jiang et al., 2021*; *Gaskin et al., 2023*; *Sukys et al., 2022*; *Tang et al., 2023a*).

In this study, we introduce DeepTX, a deep learning inference framework that integrates mechanistic models and deep learning methods to elucidate the genome-wide regulation of transcriptional bursts in DNA damage response using scRNA-seq data. The DeepTX framework comprises two modules: DeepTXsolver and DeepTXinferrer. Specifically, DeepTX initially employs the DeepTXsolver module to efficiently solve complex gene expression dynamics models using neural network architectures. It subsequently utilizes the DeepTXinferrer module to accurately infer potential transcriptional burst kinetic parameters using Bayesian methods. DeepTX demonstrates good performance on synthetic datasets. Furthermore, to investigate genome-wide regulation of transcriptional bursts in response to DNA damage, we applied the DeepTX framework to three DNA damage-related scRNA-seq datasets representing cell differentiation, apoptosis, and cell survival. As a result, we observed that the fate decision of mouse embryonic stem cells (mESCs) to undergo cell differentiation in response to DNA damage caused by 5′-iodo-2′-deoxyuridine (IdU) compounds was associated with enhanced burst size (BS enhancer) of genes linked to delayed mitosis phase transition, which may reflect regulatory programs related to reprogramming and differentiation. In colorectal cancer cells, low-dose 5-fluorouracil (5FU) treatment was associated with increased burst frequency (BF enhancer) of genes connected to intracellular oxidative stress, coinciding with apoptosis-related processes. By contrast, high-dose 5FU treatment was associated with BF enhancer genes enriched in telomerase-related pathways, which may mitigate oxidative stress and correlate with cellular resistance. In conclusion, DeepTX is a computational framework with utility and extensibility to infer transcriptional dynamics from scRNA-seq data.

## Results

### Overview of DeepTX framework

To understand how cell influences underlying transcriptional mechanisms in response to DNA damage, as described in the introduction section (*Figure 1A*), we propose an effective computational inference framework, referred to as DeepTX (*Figure 1B*). DeepTX takes as input a set of scRNA-seq datasets and then outputs genome-wide burst kinetics parameters with the benefit from deep learning methods (*Figure 1B*). The inference process of the DeepTX framework is composed of two crucial modules, DeepTXsolver and DeepTXinferrer. The first module, DeepTXsolver, is to solve for the stationary distribution of a gene expression dynamic model (*Figure 1B*, middle). A neural network is constructed in DeepTXsolver to approximate the mapping from model parameters to the corresponding stationary distribution. As in many biological contexts, there is a need to model dynamics that describe more realistic gene expression processes that are difficult to solve. For instance, gene expression under DNA damage is better represented by multi-step transcriptional models, which more faithfully reflect the underlying regulatory complexity. Moreover, sequencing noise should be integrated into the modeling framework (see Methods). The second module, DeepTXinferrer, infers the burst kinetics parameters (*Figure 1B*, middle). Using a trained neural network from DeepTXsolver, we can easily and quickly obtain the stationary distribution (called model distribution) of any parameters within a reasonable parameter space. DeepTXinferrer compares the model distribution to the stationary distribution of each gene in the scRNA-seq data (called data distribution) as the parameters are iterated until convergence and utilizes the Bayesian inference to output the posterior distribution of burst kinetic parameter (*Figure 1B*, right). We will describe these two modules in detail in the following two sections.

### DeepTX solves the transcriptional model using deep learning

To enable the recovery of kinetic information from static scRNA-seq data, we first performed rational mechanistic modeling of gene expression processes with DNA damage. The gene transcription burst process is often characterized using the traditional two-state model (*Singh et al., 2013*; *Cavallaro et al., 2021*), which assumes that genes switch randomly between OFF and ON states with

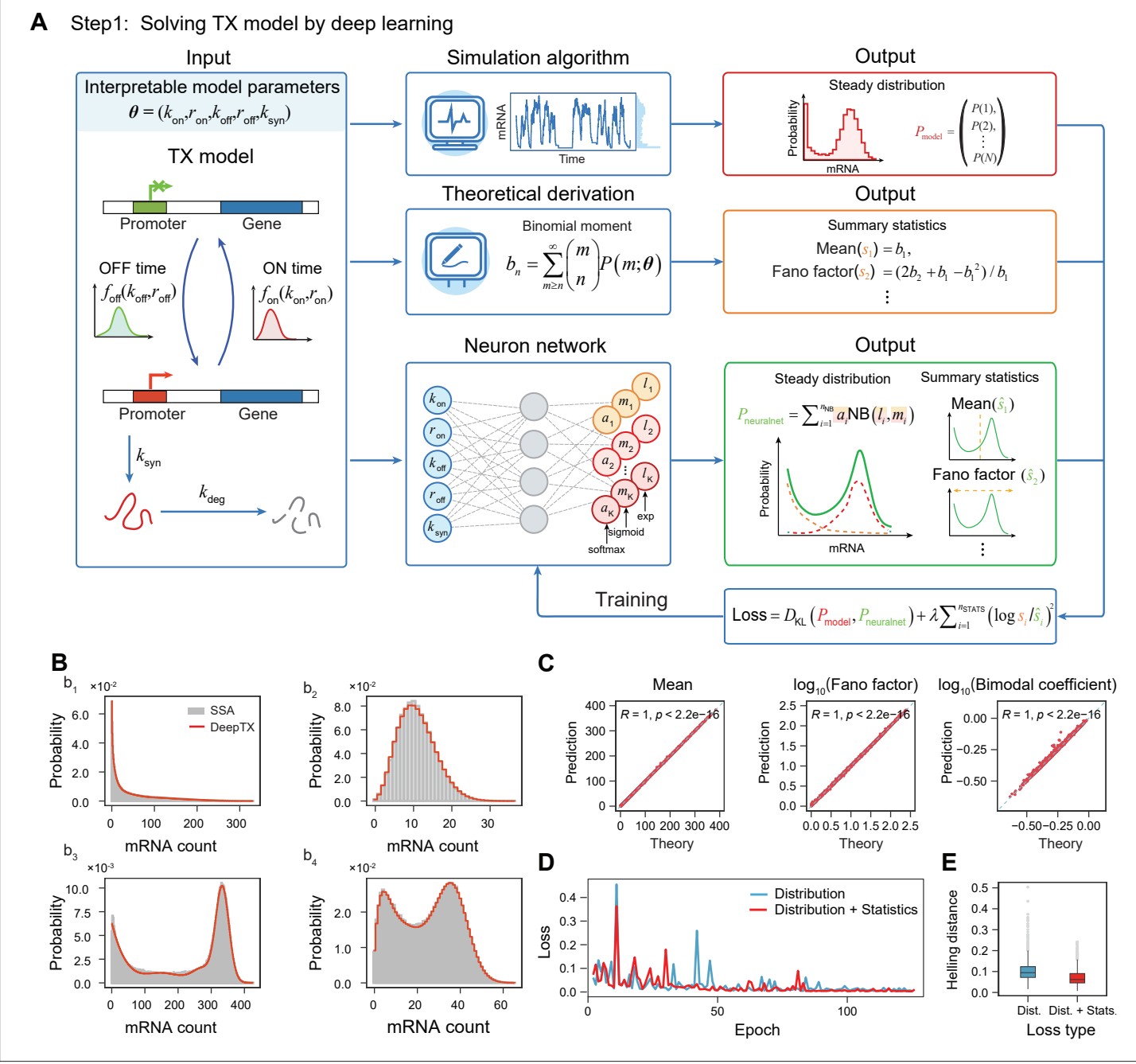

**Figure 2.** Solving the TXmodel using deep learning. (**A**) With the TXmodel and corresponding model parameters θ, we obtain a steady distribution using the stochastic simulation algorithm (SSA) simulation and the moment statistics employing binomial moment theory. These results are compared with the distribution and moment statistics of the neural network's output to calculate the neural network's loss value. The parameters of the neural network are optimized using gradient descent until convergence is achieved, indicating attainment of optimal parameters. (**B**) Comparison between SSA results and DeepTX prediction results for four sets of test parameters. The gray bars and the red stepped line correspond to the distributions obtained from SSA simulations and DeepTX predictions, respectively. (**C**) Verification results of the moment statistics predicted by DeepTX and the true moments on a test set containing 1000 elements. (**D**) The loss curve during DeepTX training. The blue curve represents that the loss function is composed of Kullback–Leibler (KL) divergence of distributions, and the red curve represents that the loss function is composed of KL divergence of distributions and statistics. (**E**) Box plots of the Hellinger distance between the true distribution and the predicted distribution by DeepTX of different loss types on the test set.

The online version of this article includes the following figure supplement(s) for figure 2:

**Figure supplement 1.** Network architecture of DeepTX.

**Figure supplement 2.** Hyperparameter tuning for DeepTX.

exponentially distributed waiting times for each state. However, the gene expression process is inherently a multi-step process, which particularly cannot be neglected under conditions of DNA damage. DNA damage can result in slowing or even stopping the RNA Pol II movement (*Lans et al., 2019*) and cause many macromolecules to be recruited for damage repair. This process will affect the spatially localized behavior of the promoter (*Lans et al., 2019*), causing the dwell time of promoter inactivation and activation that cannot be approximated by a simple two-state. We therefore used a generalized model we developed previously (*Luo et al., 2023a*), called TXmodel here, which extended the state waiting time to an arbitrary distribution, that is, the gene expression system is a non-Markovian system (*Figure 2A* and see Methods Modeling stochastic gene expression processes). More specifically, the waiting times for the transitions between the OFF and ON states are represented by two random variables, assumed to follow the arbitrary distributions $f_{\text{off}}(k_{\text{off}}, r_{\text{off}})$ and $f_{\text{on}}(k_{\text{on}}, r_{\text{on}})$ , respectively. Additionally, we assumed that the rates of mRNA synthesis and degradation are constants, that is, the waiting times for mRNA transcription and degradation follow exponential distributions with rate parameters $k_{\text{syn}}$ and $k_{\text{deg}}$ , respectively (*Figure 2A*, left). Nevertheless, this non-Markovian gene expression system is difficult to solve stationary distribution analytically, while numerical solution methods are time-consuming.

For that reason, we utilized a deep learning approach, which has been demonstrated to be effective in solving stochastic systems (*Gupta et al., 2021*), to construct a mapping from the parameter of the mechanistic TXmodel to its corresponding stationary distribution. More specifically, we aimed to train a fully connected neural network referred to as DeepTXsolver, whose inputs are the parameters of the TXmodel $\theta$ and whose output is a stationary distribution parameterized by a mixed negative binomial distribution $P_{\text{neuralnet}}$, which has good performance in approximating the solution of the chemical master equation (*Perez-Carrasco et al., 2020*; *Öcal et al., 2022*; *Figure 2*, *Figure 2—figure supplement 1* and see Methods Training neural network to solve gene expression model). To generate effective training sets for DeepTXsolver, we first generated a large number of parameter sets within a reasonable range of parameter space by using Sobol sampling (see Methods Training neural network to solve gene expression model; *Sobol', 1967*). Subsequently, for each parameter set, we employed a modified stochastic simulation algorithm (SSA) for non-Markovian TXmodels to generate numerous samples (*Figure 2A* and see Methods Training neural network to solve gene expression model). The resulting stationary distributions $P_{\text{simulation}}$ served as training labels. Additionally, we demonstrated that the arbitrary order binomial moments of the TXmodel can be analytically solved iteratively within the context of queueing theory (*Zhang et al., 2021*; *Zhang et al., 2024*). This enables the straightforward computation of significant summary statistics, including mean and Fano factor (*Figure 2A*, see Methods Modeling stochastic gene expression processes, and Appendix Model analysis). Thus, our loss function comprises two primary components: (1) the Kullback–Leibler (KL) divergence between the predicted distribution generated by the neural network $P_{\text{neuralnet}}$ and the labeled distribution generated by the SSA $P_{\text{simulation}}$, which is a widely used metric for quantifying the difference between two probability distributions; (2) the logarithmic error of the predicted summary statistics computed by the neural network $\hat{s}$ and the labeled summary statistics $s$ computed by theoretical derivation (*Figure 2A* and see Methods Training neural network to solve gene expression model). Furthermore, we systematically investigated different model architectures and hyperparameter settings to identify the configuration that achieves the best predictive performance (see Methods Training neural network to solve gene expression model and *Figure 2—figure supplement 2*).

Consequently, the loss function of DeepTXsolver converges effectively during the training process. We utilized DeepTXsolver to predict the stationary distribution using parameters from the test set. We observed that it effectively fits all four representative distributions from the TXmodel, encompassing unimodal distribution at zero point, unimodal distribution at non-zero point, bimodal distribution with one peak at zero point, and bimodal distribution with both peaks at non-zero point (*Figure 2B*). These distributions have been extensively linked to cell fate decisions in numerous experiments (*Gupta et al., 2011*; *Cohen et al., 2008*; *Bessarabova et al., 2010*). Moreover, we demonstrated that DeepTXsolver accurately predicted crucial distribution properties, including mean, Fano factor, and bimodal coefficient, with a high correlation between theoretical and predicted summary statistics ($r > 0.99$, $p$-value $< 2.2 \times 10^{-16}$, $t$-test, *Figure 2C*). It is noteworthy that, to assess the impact of the presence or absence of summary statistics on DeepTXsolver, we performed separate experiments using different loss functions. We demonstrated that incorporating summary statistics into the model

led to quicker convergence of DeepTXsolver on the training set (*Figure 2D*) and yielded more robust and accurate predictions on the test set (*Figure 2E*).

Overall, DeepTXsolver effectively establishes mappings from parameters to stationary distributions. This approach circumvents the high computational resource requirements of classical simulation algorithms and ensures the algorithm's efficiency and scalability in subsequent statistical inference, particularly in genome-wide inference.

## DeepTX infers genome-wide transcriptional kinetics from scRNA-seq data

Having obtained the trained neural network model, we can rapidly compute the stationary distributions of any TXmodel parameters. These distributions represent the outcomes of the underlying true gene expression process, referred to as the 'true distribution' $P_{model}$ (*Figure 3A*). However, during the inference process, the gene expression distributions measured by the scRNA-seq data we utilized, termed the 'observed distribution of the data' $P_{data}$, were subject to noise, including errors stemming from the measurement technique (*Sarkar and Stephens, 2021*; *Figure 3A*). Thus, we introduced a mechanistic hierarchical model to bridge the gap between the true and observed distributions. This hierarchical model considers that the observed data results from the interplay between the gene expression process and the measurement process. It combines the two distributions using a convolutional form $P_{data}(Y = y) = \int_0^\infty P_{measure}(y|x)P_{model}(x)dx$ (see Methods Inferring burst kinetics from scRNA-seq data with DeepTXinferrer). $P_{model}(x)$ represents the mixed negative binomial distribution approximated, whereas for $P_{measure}(y|x)$, we opted for the binomial distribution. This choice is informed by the fact that it can approximate hypergeometric distributions representing sequencing sampling without replacement when the sample size is sufficiently large. Consequently, the final distribution can maintain the form of a mixed negative binomial distribution (Appendix Derivation of hierarchical negative binomial mixture distribution), denoted as the observed distribution of the model $P_{data}$ (*Figure 3A*).

Leveraging the hierarchical model, we can construct a scalable and interpretable module called DeepTXinferrer, facilitating genome-wide inference of transcriptional burst kinetics from scRNA-seq data. First, we input the prior distribution of mechanistic model parameters and a set of scRNA-seq data, and we can get the observed distribution of the model $P_{obsmodel}$ (derived successively from the trained neural network and the hierarchical model) and the observed distribution of the data $P_{obsdata}$, respectively (*Figure 3A*). Second, we used the Hellinger distance $L(\theta) = D(P_{obsmodel}, P_{obsdata})$ to measure the difference between $P_{obsmodel}$ and $P_{obsdata}$ (see Methods Inferring burst kinetics from scRNA-seq data with DeepTXinferrer). It is worth noting that the loss function remains solvable via gradients, despite the inclusion of complex computations, such as neural networks and hierarchical models. Specifically, to avoid the gradient-based optimization process from getting trapped in local optima, we employ a black-box optimization approach to select the initial point for parameter inference (*Das and Suganthan, 2010*). Consequently, we utilize a gradient-based optimizer to update the iterative parameters until convergence. Lastly, we employ a method based on loss potentials, which computes the posterior distribution of the parameters during the optimization of the loss function (*Gaskin et al., 2023*; *Figure 3A*, see Methods Inferring burst kinetics from scRNA-seq data with DeepTXinferrer). Subsequently, this process yields results of the transcriptional burst kinetics derived from the estimated parameters of the mechanistic model.

We assessed the effectiveness of DeepTXinferrer by conducting validation on synthetic data (see Methods Inferring burst kinetics from scRNA-seq data with DeepTXinferrer). Synthetic samples were generated for each parameter set, and our inference algorithm was then applied to derive the output transcriptional burst kinetics (see Methods Inferring burst kinetics from scRNA-seq data with DeepTXinferrer). As a result, the estimation of transcriptional burst kinetics demonstrated high accuracy and strong correlation across most parameter regions ($r > 0.99$, $p$-value $< 2.2 \times 10^{-16}$ of $t$-test; *Figure 3—figure supplement 1*), particularly within the region of fully expressed mRNA abundance (*Figure 3B*). Moreover, the estimated posterior distribution of each parameter suggests that the peak value closely approximates the true value (*Figure 3C–E*; *Figure 3—figure supplement 2*). Additionally, we verified the robustness of our algorithm concerning the number of samples, observing that higher sample numbers corresponded to increased accuracy in estimation (*Figure 3—figure supplement 1*). We additionally examined the stability of inferred burst phenotypes under varying capture efficiencies and found consistently strong correlations between inferred burst size and BF across capture efficiency

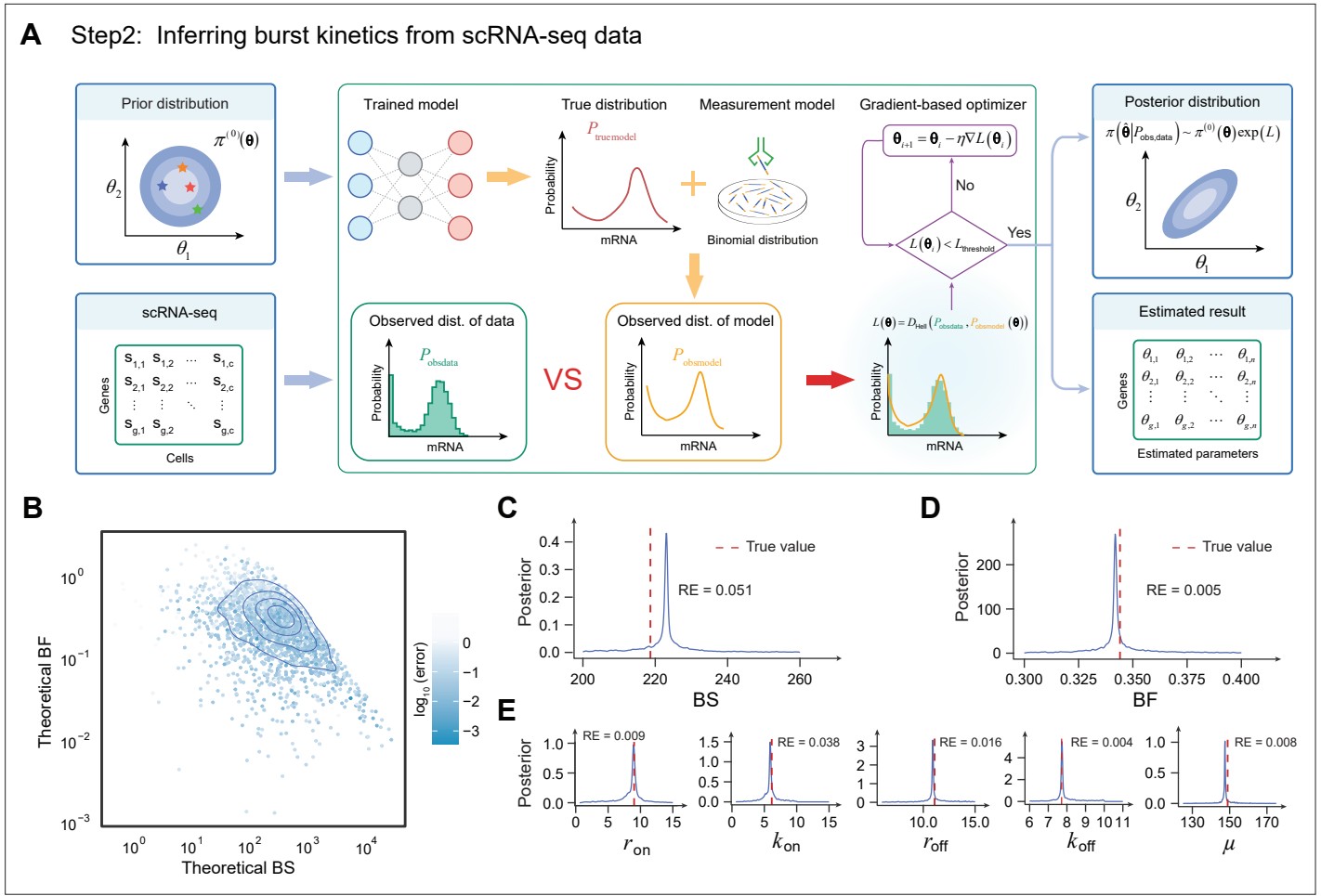

**Figure 3.** Inferring TXmodel from scRNA-seq data. (**A**) The dynamic parameters are sampled from a given prior distribution, and as input to the neural network, the solution of the corresponding dynamic model can be obtained by using this parameter. The solution of the dynamic model is mixed with the binomial distribution to obtain the model observed distribution. Loss values were obtained using the observed distribution of the data obtained from the scRNA-seq data compared to the observed distribution of the model. The loss values are optimized until convergence using gradient descent to obtain the parameters of the mechanism model and the posterior distribution of its parameters. (**B**) The scatters represent the real BS and BF values of the stochastic simulation algorithm (SSA) synthetic set parameters, and the depth of the color represents the error between the inferred BS (BF) and the true BS (BF). (**C, D**) The blue solid line represents the edge density of the burst kinetics of the model, and the red dashed line represents the true value. (**E**) The marginal density for the five parameters of the model, where the red dashed line represents the true value. The relative error (RE) quantifies the discrepancy between the true parameter and the peak of the inferred posterior distribution.

The online version of this article includes the following figure supplement(s) for figure 3:

**Figure supplement 1.** Precision and robustness of inferred results.

(**a**) Verification of the Hellinger distance between the distribution obtained by DeepTXsolver and the distribution obtained by stochastic simulation algorithm (SSA) simulation of the test set. (**b, c**) Correlation scatterplot of inferred and theoretical BS (BF). (**d, e**) Box plots of BF and BS obtained by inferring from different cell numbers, where the red dashed line corresponds to the true parameter.

**Figure supplement 2.** Posterior marginal distributions of TXmodel parameters under an alternative setting.

**Figure supplement 3.** Comparison of inferred burst dynamics under different capture efficiencies.

**Figure supplement 4.** Comparison of inference accuracy and efficiency between DeepTX and other algorithms.

settings (*Figure 3—figure supplement 3*). Compared to previous work (*Larsson et al., 2019*; *Luo et al., 2023a*; *Tang et al., 2023b*; *Gu et al., 2025*), DeepTX demonstrates superior performance in both efficiency and accuracy (*Figure 3—figure supplement 4*). In summary, our inference algorithm exhibited strong performance on synthetic data, affirming its reliability for application to real-world data in inferring transcriptional bursts.

In the subsequent sections, we will employ the DeepTXinferrer to explore how cells determine diverse fates in response to DNA damage stimuli through genome-wide regulation of transcriptional bursting. We analyzed three sets of scRNA-seq data representing distinct cell fates: cell differentiation, apoptosis, and survival. The first set of data is a mutually controlled scRNA-seq dataset of mESC cell differentiation induced by IdU drug treatment. This dataset contains 12,481 genes of transcriptomes from 812 cells and 13,780 genes of transcriptomes from 744 cells, respectively. The second set of data is a mutually controlled scRNA-seq dataset of human colon cancer cells treated with low-dose 5FU drug that causes apoptosis. This dataset contains 8534 genes of transcriptomes from 1673 cells and 7077 genes of transcriptomes from 632 cells, respectively. The third set of data is a mutually controlled scRNA-seq dataset of human colon cancer cells in which high-dose 5FU drug treatment resulted in cell-resistant survival. This dataset contains 8534 genes of transcriptomes from 1673 cells and 6661 genes of transcriptomes from 619 cells, respectively.

## DeepTX reveals burst size enhancement induced by IdU treatment associated with cell differentiation

The administration of the IdU drug, a thymine analogue, influences the gene expression process within cells, subsequently impacting cell differentiation (*Desai et al., 2021*; *Li et al., 2024*). The perturbation in the gene expression process primarily stems from DNA damage, as IdU is randomly integrated into the DNA chain, thereby inducing DNA damage (*Li et al., 2024*). Although DNA damage caused by IdU treatment has little impact on genome-wide mean gene expression, it increases variance across the genome, thereby affecting cell differentiation (*Desai et al., 2021*). This is different from the understanding that increases in variance are usually caused by fluctuations in the mean gene expression (*Newman et al., 2006*). Meanwhile, the mean can be deconvolved as the product of BS and BF. This raises the question of the relationship between bursting dynamics and variance on a genome-wide scale, and how changes in bursting dynamics affect cell differentiation.

We used the DeepTX framework to infer a set of preprocessed scRNA-seq data (see Methods IdU and DMSO dataset) to obtain underlying burst kinetics (*Figure 4*). The difference between the statistics and distribution obtained by inference and the statistics and distribution of sequencing data is minimal, ensuring the correctness of the inference results (*Figure 4—figure supplement 1*). Subsequently, we observed considerable variations in BS and BF across different genes, with some genes exhibiting minimal changes in mean expression but pronounced differences in variance (*Figure 4*, *Figure 4—figure supplement 2a–f*). Notably, genes with significantly increased BS exhibit corresponding decreases in BF, suggesting relative stability in the average gene expression level (*Figure 4—figure supplement 2d-f*). Additionally, we observed a significant increase in transcriptional BS for most genes in the treatment group (*Figure 4A*). This indicates that the rise in variance primarily results from the enhanced BS induced by DNA damage (*Figure 4B*). Notably, this conclusion is consistent with the results of theoretical analysis (Appendix 2). To further explore the biological characteristics of BS enhancement, we identified genes with upregulated BS using differential analysis methods. Further, we perform GO enrichment analysis (*The Gene Ontology Consortium, 2015*) on the identified genes. We observed that the genes with enhanced BS revealed significant enrichment in terms related to mitotic cell cycle checkpoint signaling, alongside pathways involved in DNA damage response and cell cycle transitions (*Smits and Medema, 2001*; *Figure 4D*). Specifically, the mitotic checkpoint serves as a crucial safeguard to ensure accurate chromosome segregation and maintain genomic stability under DNA damage conditions. Activation of the mitotic checkpoint can influence cell fate decisions and differentiation potential. Sustained activation of the spindle assembly checkpoint has been reported to induce mitotic slippage and polyploidization, which in turn may enhance the differentiation potential of embryonic stem cells. Subsequently, we performed Gene Set Enrichment Analysis (GSEA) (*Subramanian et al., 2005*) on the identified differentially expressed genes associated with BS and also observed enrichment in pathways related to cell cycle transitions and mitotic intra-S DNA damage checkpoint signaling (*Figure 4E, F*). Similarly, we subjected genes significantly enhanced by BF in the treatment group (*Figure 4—figure supplement 2f*) to GO analysis (*Figure 4—figure supplement 2g*). While it is tempting to hypothesize that enhanced BS may contribute to DNA damage-related checkpoint activation and thereby influence cell cycle progression and differentiation, our current results only indicate an association between burst size enhancement and pathways involved in DNA damage response and checkpoint signaling. This is consistent with

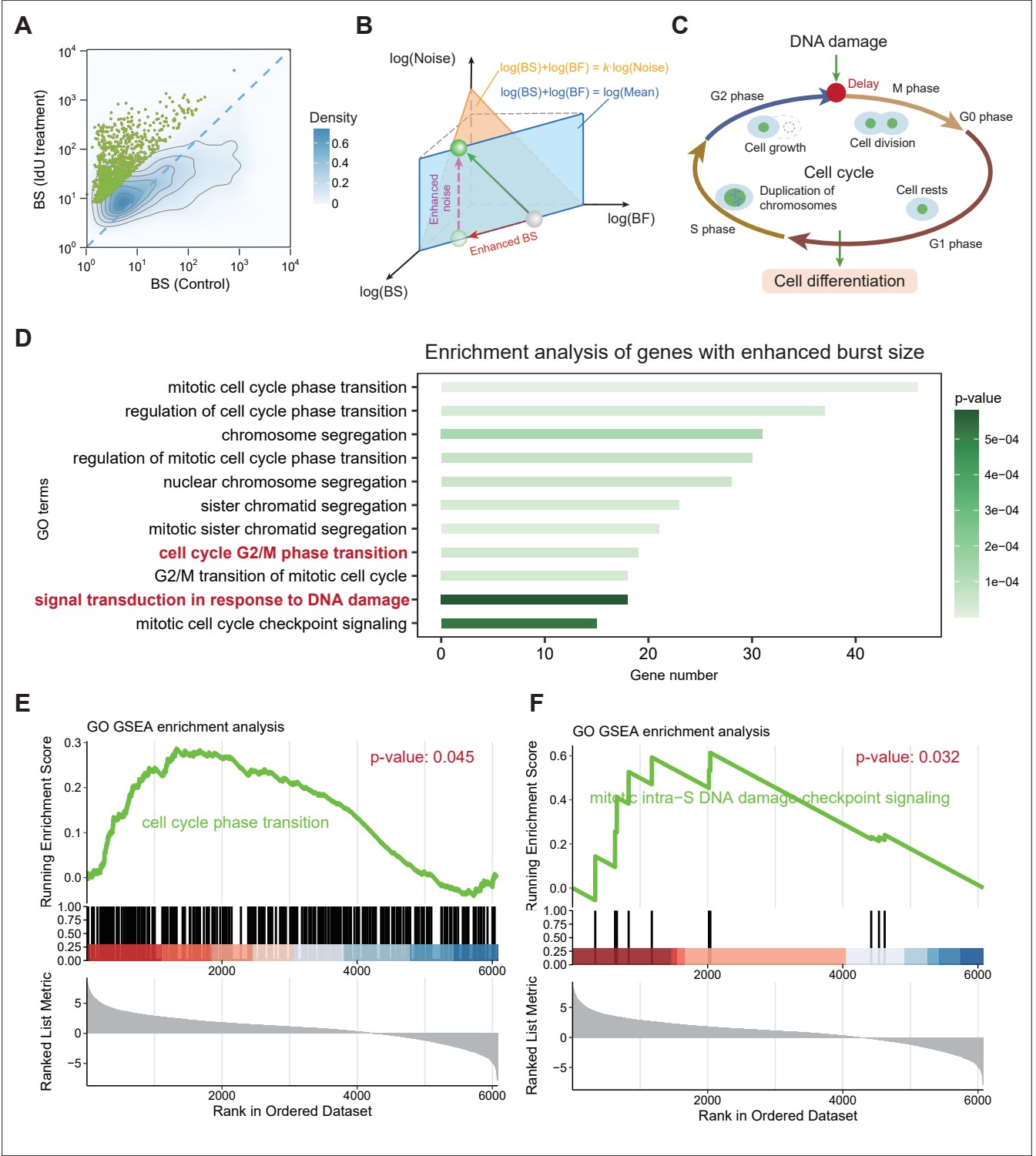

**Figure 4.** Burst size enhancement is linked to a delay in the cell cycle. (**A**) Density map of BS of genes treated with 5'-iodo-2'-deoxyuridine (IdU) drugs compared to BS of genes not treated with drugs. Green dots represent upregulated BS differential genes. (**B**) Schematic representation of the simultaneous enhancement of both BS and noise of a gene accompanied by the mean value of the gene remaining unchanged after IdU drug treatment. In the log–log–log 3D space, the mean expression level lies on a blue plane defined by $\log(\text{BS}) + \log(\text{BF}) = \log(\text{Mean})$, indicating that it is determined by the product of burst size and burst frequency. The orange plane captures a scaling relationship between expression noise and burst kinetics, described by $\log(\text{BS}) + \log(\text{BF}) = \kappa \log(\text{Noise})$, where $\kappa$ is a constant reflecting their covariation. Notably, the trajectory of the green sphere shows that, under a fixed mean expression level (i.e., confined to the blue plane), increased gene expression noise primarily results from an increase in

*Figure 4 continued on next page*

*Figure 4 continued*

burst size. (**C**) The schematic diagram illustrates the hypothesis that DNA damage induced by IdU treatment may affect gene transcription, potentially influencing cell cycle transitions during mitosis and regulating cellular differentiation. (**D**) Gene GO enrichment analysis is performed on the green dots of BS in (**A**) to obtain enrichment pathway diagram. The bold red pathways highlight the key components involved in the cellular differentiation mechanism. (**E, F**) Pathways obtained by GSEA enrichment analysis of BS of genes.

The online version of this article includes the following figure supplement(s) for figure 4:

**Figure supplement 1.** Inference results of scRNA-seq data treated with DMSO and 5'-iodo-2'-deoxyuridine (IdU).

**Figure supplement 2.** Analysis results of scRNA-seq data treated with 5'-iodo-2'-deoxyuridine (IdU) and DMSO.

previous experimental observations showing that IdU treatment induces DNA damage and alters cell fate decisions through effects on the cell cycle (*Rosina et al., 2019*; *Riccio et al., 2022*; *Liu et al., 2017*; *Figure 4C*). Additionally, research indicates that enhanced cell differentiation results from delayed cell cycle transitions (*Rosina et al., 2019*).

## DeepTX reveals BF enhancement under low-dose 5FU treatment and its association with apoptosis

The chemotherapeutic compound, 5FU, exerts its cytotoxic effects on colorectal cancer cells by inflicting damage to their DNA (*Kuipers et al., 2015*; *Bunz et al., 1999*; *Chang et al., 2014*). This damage triggers apoptosis, a process of programmed cell death, thus inhibiting the growth and proliferation of malignant cells (*Kuipers et al., 2015*; *Bunz et al., 1999*; *Chang et al., 2014*). However, the impact of 5FU-induced DNA damage on the gene expression process and its potential relationship to cell apoptosis remains incompletely understood. In this section, we employ the DeepTX to investigate the effect of 5FU-induced DNA damage on the gene expression process in colorectal cancer cells with scRNA-seq data (*Park et al., 2020*). First, we inferred the burst kinetics from the scRNA-seq data of cells treated with low doses (10 μM) of 5FU and controls (*Figure 5*). After preprocessing the data (see Methods 5FU dataset), we inferred the scRNA-seq data to obtain a distribution that fits the scRNA-seq data (*Figure 5—figure supplement 1*), as well as the underlying burst kinetics (*Figure 5A, B*). Further, we found that the mean difference between the two groups of scRNA-seq data was not significant ($p$-value = 0.5 of $t$-test), but the variance and BS were significantly different ($p$-value < 0.05 of $t$-test; *Figure 5A*, *Figure 5—figure supplement 2a–c*). Then, we performed a differential analysis of the BS and BF of gene expression between the control cells and the cells treated with 10 μM 5FU, and then performed GO gene enrichment analysis on differential genes (*Figure 5C*, *Figure 5—figure supplement 2d–f*). We found that BS downregulated differential genes were mainly enriched in apoptosis pathways (*Hongmei, 2012*), electron exchange pathways, and oxidation-related pathways (*Figure 5E*, *Figure 5—figure supplement 2g*). Specifically, the enrichment analysis indicated that although the burst size of genes in 5FU-treated cells was generally larger than that in the control group, the genes primarily influencing cell apoptosis exhibited a downregulated BS.

These enrichment results are in line with previously reported biological processes (*Figure 5D*). First, studies in recent years have found that many anticancer drugs trigger apoptosis and necrosis of cancer cells by activating the production of reactive oxygen species (ROS) (*Sun and Rigas, 2008*). And experiments have shown that 5FU can induce ROS production in colorectal cancer cells (oxidation-related pathways) (*Laha et al., 2015*; *Bwatanglang et al., 2016*; *Chenna et al., 2022*). Meanwhile, mitochondria are the main source of ROS production, and defects in the mitochondrial electron transport chain will increase ROS production (electron exchange pathways) (*Adam-Vizi, 2005*). As a result, excessive production of ROS can cause mitochondrial DNA damage and nuclear DNA damage (*Figure 5D*). In particular, when mitochondrial DNA is damaged, mitochondrial damage repair will further increase the pressure of ROS (*Figure 5D*; *Shokolenko et al., 2014*). Overall, our results support an association between 5FU-induced burst kinetics changes and pathways implicated in oxidative stress and apoptosis (*Figure 5D*; *Handali et al., 2018*; *Hwang et al., 2001*).

## DeepTX reveals BF enhancement under high-dose 5FU treatment and its association with survival pathways

Low-dose 5FU treatment can induce cell apoptosis, while research shows high-dose 5FU treatment can induce cell resistance and evade apoptosis to survive (*Kuranaga et al., 2001*). Hence, this part

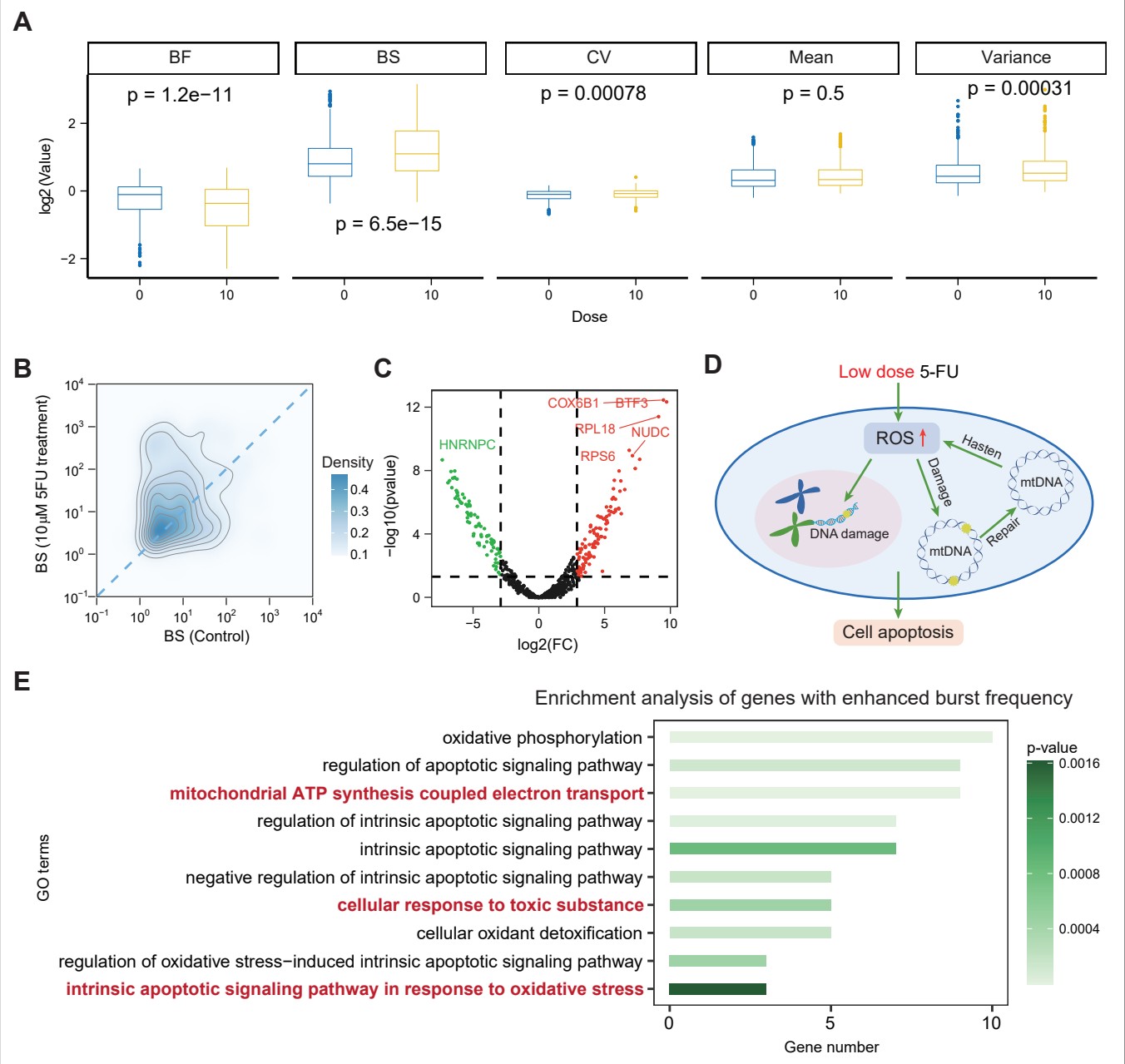

**Figure 5.** Low-dose 5-fluorouracil (5FU) treatment is linked to increased burst frequency and cell apoptosis. (**A**) Comparison of gene bursting kinetics and statistics between control and 10-dose 5FU treated cells. (**B**) Density map of inferred burst size. (**C**) Volcano map of the fold changes (FC) of inferred burst size between control and 10-dose 5FU treated cells. (**D**) The diagram hypothesizing the potential association between low-dose 5FU treatment and the induction of apoptosis. (**E**) The enrichment analysis results were derived from genes that were downregulated in BS and upregulated in BF in the experimental group. The bold red pathways highlight the key components involved in the cellular apoptotic mechanism.

The online version of this article includes the following figure supplement(s) for figure 5:

**Figure supplement 1.** Inference results of scRNA-seq data treated with low-dose 5-fluorouracil (5FU) and without 5FU treatment.

**Figure supplement 2.** Analysis results of scRNA-seq data treated with low-dose 5-fluorouracil (5FU) and without 5FU treatment.

mainly studies the impact of differences in burst kinetics of gene expression between high-dose (50 µM) 5FU-treated and control colorectal cancer cells on cell fate decisions.

First, we inferred the burst kinetics from the scRNA-seq data of cells treated with high doses (50 µM) of 5FU and controls (*Figure 6*). After data preprocessing (see Methods 5FU dataset), we inferred the scRNA-seq data to obtain the distribution that fits the scRNA-seq data (*Figure 6—figure supplement*

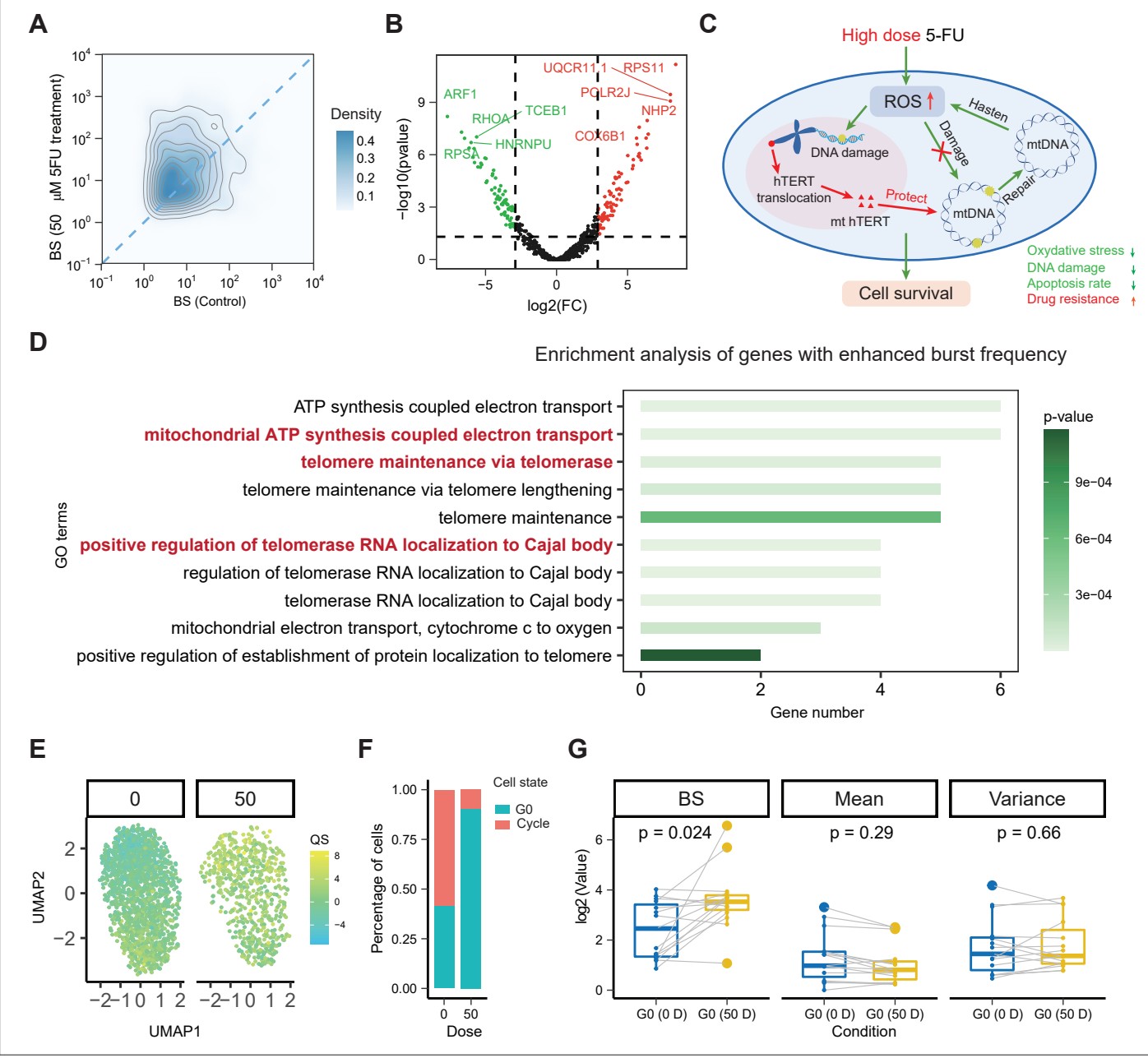

**Figure 6.** High-dose 5-fluorouracil (5FU) treatment may promote cell drug resistance by altering burst frequency. (**A**) Density map of inferred burst size. (**B**) Volcano map of the fold changes (FC) of inferred burst size between control and 50-dose 5FU treated cells. (**C**) Schematic diagram hypothesizing the mechanism by which 5FU drug treatment may induce cells to acquire antioxidant properties, potentially enabling continued survival. (**D**) The enrichment analysis results were derived from genes that were downregulated in BS and upregulated in BF in the experimental group. (**E**) UMAP of G0 arrest quality score (QS) of control and 50-dose 5FU treated cells. (**F**) Proportion of G0 arrested (drug-resistant) and cycling (non-resistant) cells. (**G**) Comparison of gene bursting kinetics and statistics of G0 arrested genes between control and 50-dose 5FU treated cells. The light gray lines indicate the trend of changes in the corresponding bursting kinetics values, mean, and variance for each gene.

The online version of this article includes the following figure supplement(s) for figure 6:

**Figure supplement 1.** Inference results of scRNA-seq data treated with high-dose 5-fluorouracil (5FU) and without 5FU treatment.

**Figure supplement 2.** Analysis results of scRNA-seq data treated with high-dose 5-fluorouracil (5FU) and without 5FU treatment.

1), as well as the underlying burst kinetics (*Figure 6A*). We similarly observed the phenomenon that the average expression value of genes remained stable, while there was an enhanced increase in the genome-wide variance (*Figure 6—figure supplement 2a–c*). Further, we performed a difference analysis on the BS of the two sets of inferred data and selected genes with large differences in BS and BF for GO gene enrichment (*Figure 6A, B*, *Figure 6—figure supplement 2d–f*). We found that the BS downregulated differentially expressed genes were mainly enriched in the Cajal body-related pathway, the electron exchange pathway, and telomerase-related pathways (*Nabetani and Ishikawa, 2011*), but the apoptosis-related pathways were not significant (*Figure 6D*). In particular, according to experimental studies, telomerase-related pathways can activate telomerase and play an important role in the elongation of telomeres (*Cristofari et al., 2007*). These enrichment results suggest that high-dose 5FU treatment may be associated with processes such as telomerase activation and mitochondrial function maintenance, both of which have been implicated in cell survival and apoptosis evasion in previous experimental studies. For example, prior work indicates that Human Telomerase Reverse Transcriptase (hTERT) translocation can activate telomerase pathways to support telomere maintenance and reduce oxidative stress, which is thought to contribute to apoptosis resistance (*Kuranaga et al., 2001*). Overall, the observed transcriptional bursting changes are consistent with these reported survival-associated mechanisms.

Gene enrichment results are consistent with existing ways of cells evading apoptosis (*Figure 6C*). That is, endogenous ROS production exceeds the cellular antioxidant defense capacity, leading to chemical damage to mitochondrial DNA. Meanwhile, high doses of 5FU cause hTERT translocation, which on the one hand triggers telomerase activity and maintenance pathways and enables cells to evade apoptosis (*Kuranaga et al., 2001*). On the other hand, this translocation will enrich hTERT in mitochondrial DNA, thereby reducing oxidative stress and mitochondrial DNA damage to protect mitochondria and protect cells from apoptosis (*Lipinska et al., 2017*).

Resistance to therapy has also been associated with G0 arrest, a non-proliferative state characteristic of persister cells (*Wiecek et al., 2023*). To verify the presence of resistant cells in cells treated with high doses of 5FU, we selected genes related to G0 arrest and performed gene set variation analysis (*Hänzelmann et al., 2013*) to obtain the G0 arrest quality score (*Wiecek et al., 2023*) of each cell and perform UMAP dimensionality reduction visualization (*Figure 6E*). Through arrest quality score discrimination, we found that the proportion of resistant cells was high among cells treated with high doses of 5FU (*Figure 6F*). This result was consistent with the results of our enrichment analysis. Specifically, by comparing the BS and mean values of these genes, we found that most of the BS of G0-arrest-related genes were upregulated, but the mean changes were small (*Figure 6G*).

## Discussion

Gene expression is inherently stochastic, and transcriptional bursting contributes to inter-individual variability in gene expression patterns (*Dar et al., 2012*; *Zenklusen et al., 2008*; *Phillips et al., 2012*), although it represents only one of multiple sources of cellular heterogeneity. Experimental observations confirm this bursting mechanism as a cellular response to environmental changes, including DNA damage (*Friedrich et al., 2019*; *Desai et al., 2021*), but its precise role in DNA damage response and subsequent cell fate decisions remains an emerging hypothesis. In this study, we introduced a mechanism-based deep learning framework (DeepTX) for performing genome-wide inference on transcriptional burst kinetics of a comprehensive gene expression model using scRNA-seq data. DeepTX provides an efficient computational approach to characterize burst kinetics across genes, enabling the generation of testable hypotheses about how transcriptional bursting may be associated with DNA damage response and potential changes in cellular state.

Utilizing DeepTX, we analyzed three sets of scRNA-seq data comprising control and stimulus pairs driving different cell fate decisions: cell differentiation, apoptosis, and survival. This analysis unveiled significant insights into the underlying mechanisms of transcriptional burst kinetics. First, the IdU drug treatment-induced enhancement of BS in genes appears to be associated with a delayed transition in the cell mitosis phase, which may in turn be related to changes in cell reprogramming and differentiation. Interestingly, while prior studies have suggested that noise enhancement contributes to cell fate decisions, our observations indicate that BS enhancement may reflect a deeper regulatory feature correlated with cell differentiation. Second, low-dose 5FU treatment is associated with increased oxidative stress and an accompanying elevation in BF, which may relate to enhanced apoptotic

activity. Consistent with previous findings linking apoptosis to ROS stress, our study suggests that ROS-induced DNA damage is correlated with changes in gene expression burst kinetics, which may contribute to the apoptotic outcomes observed. Third, high-dose 5FU treatment induces telomerase extension, mitigating oxidative stress and fostering drug resistance by upregulating BF. Despite prior evidence of drug-resistant cells, our enrichment analysis underscores telomerase extension's role in resisting oxidative stress, underscoring the impact of BF enhancement.

We highlight several advantages of our DeepTX framework, which differs significantly from previous work that utilizes scRNA-seq data to study diverse noise sources from several aspects (*Eling et al., 2019*; *Faure et al., 2017*; *Morgan and Marioni, 2018*; *Ochiai et al., 2020*). First, DeepTX employs a mechanistic hierarchical model that is both interpretable and extensible. To enhance the model's realism in interpreting scRNA-seq data, we developed a novel approach integrating the sequencing process and the underlying gene expression process. Specifically, we employed the TXmodel, a complex non-Markovian stochastic dynamic model that extends the waiting times for gene OFF and ON states into a non-exponential distribution, reflecting the multi-step nature of gene regulation. As a result, TXmodel not only captures gene expression dynamics under DNA damage but also depicts other multi-step regulatory processes, such as chromatin opening (*Lomvardas and Thanos, 2002*), preinitiation complex formation (*Fuda et al., 2009*), transcription factors binding (*Nordick et al., 2022*). Second, despite the mathematical complexity of the mechanistic hierarchical model, DeepTX renders it tractable. This is achieved by (1) directly mapping model parameters to corresponding stationary distributions via a neural network trained on extensive simulated single-cell RNA data and theoretical analysis summary statistics for TXmodel; (2) approximating the stationary distribution output of TXmodel, coupled with the sequencing process, by a mixed negative binomial distribution, maintaining the final distribution format. Third, DeepTX's inference process is scalable, allowing genome-wide bursting kinetics inference from static scRNA-seq data. Despite the computational complexity of parameter inference involving hierarchical models and trained neural networks, DeepTX retains auto-differentiation for efficient and accurate gradient computation.

The DeepTX research paradigm contributes to a growing line of area aiming to link mechanistic models of gene regulation with scRNA-seq data. Maizels provided a comprehensive review of computational strategies for incorporating dynamic mechanisms into single-cell transcriptomics (*Maizels, 2024*). In this context, RNA velocity is one of the most important examples as it infers short-term transcriptional trends based on splicing kinetics and deterministic ODE models. However, such approaches are limited by their deterministic assumptions and cannot fully capture the stochastic nature of gene regulation. DeepTX can be viewed as an extension of this framework to stochastic modeling, explicitly addressing transcriptional bursting kinetics under DNA damage. Similarly, DeepCycle, developed by Sukys and Grima (*Sukys and Grima, 2025*) investigates transcriptional burst kinetics during the cell cycle, employing a stochastic age-dependent model and a neural network to infer burst parameters while correcting for measurement noise. By contrast, MIGNON integrates genomic variation data and static transcriptomic measurements into a mechanistic pathway model (HiPathia) to infer pathway-level activity changes, rather than gene-level stochastic transcriptional dynamics (*Garrido-Rodriguez et al., 2021*). In this sense, DeepTX and MIGNON are complementary, with DeepTX resolving burst kinetics at the single-gene level and MIGNON emphasizing pathway responses to genomic perturbations, which could inspire future extensions of DeepTX that incorporate sequence-level information.

The DeepTX research paradigm is poised for extension to encompass both dynamic models and biological data. On the one hand, our focus is currently confined to the transcriptional processes influenced by DNA damage, limiting the explanatory scope of the TXmodel to larger-scale biological phenomena such as alternative splicing (*Wan et al., 2021*; *Gorin and Pachter, 2022*), protein translation (*Luo et al., 2023a*; *Golan-Lavi et al., 2017*), epigenetic modification (*Nicolas et al., 2018*), and chromatin movement (*Liu et al., 2016*; *Wang et al., 2024*; *Wang et al., 2023*). Moreover, further development is needed to incorporate additional regulatory factors, including signaling pathways (*Garrido-Rodriguez et al., 2021*), cell cycle progression, cell size (*Sukys and Grima, 2025*), and leaky transcription (*Kepler and Elston, 2001*). These broader biological processes necessitate more intricate dynamic models, often challenging to solve and thus challenging to apply to realistic biological data.

On the other hand, inference from static scRNA-seq relies on the steady-state assumption, which has some limitations. First, in some scenarios, the cell system may exhibit highly transient

transcriptional programs that do not satisfy stationarity, leading to biased or misleading parameter estimates. For example, immediately following a synchronized developmental stimulus such as serum shock-induced activation of immediate-early genes, transcription levels increase significantly (*Hope et al., 1992*). Second, because DeepTX infers the mean BF and size across the population, it cannot recover the underlying time-resolved dynamics or distinguish heterogeneous kinetic subpopulations. Third, lack of time-resolved measurements may affect the accuracy of inferences about dynamic parameters, especially the unidentifiability of parameters inferred from steady-state distributions, that is, multiple parameters leading to the same steady-state distribution. The unidentifiability of individual parameters is a common and critical problem in systems biology studies. Hence, in DeepTX, we employed a Bayesian approach based on loss potential to infer the posterior distributions of the parameters. Although DeepTX also encounters the issue of unidentifiability for individual parameters (*Figure 3—figure supplement 2*), the multimodal nature of the posterior distribution suggests that multiple distinct parameter sets can produce similarly good fits to the observed data, highlighting the inherent non-identifiability of the model. Nevertheless, in the multimodal posterior distribution, at least one of the posterior peaks aligns closely with the ground truth, thereby demonstrating the validity of the inferred result. In future work, integrating time-series single-cell measurements with other emerging data types could help overcome these limitations. Examples include increasingly comprehensive transcriptome data with both temporal and spatial resolution (*Wen and Tang, 2022*), as well as state-of-the-art multi-omics data (*Maizels, 2024*; *Miao et al., 2021*). Moreover, another limitation of the current implementation is that DeepTX is only trained on simulated datasets. While providing controlled benchmarks for mechanistic validation, it may not fully recapitulate the heterogeneity and stochastic complexity of real biological systems. In the future work, publicly available annotated scRNA-seq datasets could be used to complement this simulation-based training strategy and enhance generalizability.

In conclusion, DeepTX provides a step toward inferring complex transcriptional kinetics from scRNA-seq data through mechanism-based deep learning inference. It sheds light on the augmented burst kinetics in DNA damage response across various cell fate decisions, thereby offering novel insights.

## Methods
### The DeepTX framework
#### Framework overview
The DeepTX framework aims to infer the underlying bursting dynamics from static scRNA-seq data to explore the impact of changes in bursting dynamics on cell fate decisions. The DeepTX framework includes two modules, DeepTXsolver and DeepTXinferrer. DeepTXsolver can efficiently and accurately solve transcription models using neural network architecture, and DeepTXinferrer can infer scRNA-seq data to obtain its underlying transcriptional burst kinetics using Bayesian methods.

The input of the DeepTXsolver is the parameters of the mechanism model, and the output is the corresponding stationary distribution solution of the model. The input to the DeepTXinferrer is the discrete probability distribution of gene expression obtained from scRNA-seq data, and the output of the model is the parameters of the mechanism model.

The construction of the DeepTX framework consists of the following parts: (1) Model the gene expression process. (2) Train a neural network that solves the mechanism model using the dataset generated by SSA simulation. (3) Hierarchical model modeling to infer underlying burst kinetic parameters from scRNA-seq data.

When building the DeepTX Framework, we made the following assumptions: (1) The gene expression of cells was in a stationary distribution during sequencing. This assumption has been widely adopted in scRNA-seq studies, as it effectively captures mRNA expression distributions and enables the inference of underlying dynamic parameters (*Larsson et al., 2019*; *Luo et al., 2023a*; *Ramsköld et al., 2024*; *Gupta et al., 2022*). (2) Gene expression counts of the same cell type follow the same distribution. Given that cells of the same type typically share similar transcriptional programs, their gene expression distributions are approximately identical, supporting the validity of this commonly used assumption. (3) The solution of the model can be approximated by a mixed negative binomial distribution. Since the stationary solution of the chemical master equation can be expressed as a

Poisson mixture, which can be more efficiently approximated using negative binomial mixtures (*Gorin et al., 2024*). (4) The state space sampled from a sufficiently long single simulation is statistically equivalent to that obtained from multiple simulations at steady state in gene expression models (*Kuntz et al., 2021*).

## Modeling stochastic gene expression processes

To recover dynamic transcriptional burst kinetic parameters from static scRNA-seq data, we require the stochastic modeling approach to profile the gene expression process. The traditional telegraph model is often used to model gene transcriptional bursting, which assumes that switching between genes follows an exponential distribution (i.e., switching is a single-step process) (*Peccoud and Ycart, 1995*; *Kim and Marioni, 2013*). However, this assumption is an oversimplification of the gene expression process in the context of DNA damage. Due to the fact that DNA damage slows down and even stops the elongation of RNA Pol II across the double-strand DNA (*Lans et al., 2019*), the gene expression process should be regarded as a multi-step process. This property causes the waiting time for the gene switching between active and inactive states to follow a non-exponential distribution (*Soltani et al., 2015*; *Zhang and Zhou, 2019a*). Therefore, we use TXmodel (*Luo et al., 2023a*) to better characterize the gene transcription process in the context of DNA damage (*Figure 2*), whose chemical reaction network (CRN) can be written as follows:

$$R_1 : \text{OFF} \xrightarrow{f_{\text{off}}(t,\boldsymbol{\omega}_{\text{off}})} \text{ON}$$

$$R_2 : \text{ON} \xrightarrow{f_{\text{on}}(t,\boldsymbol{\omega}_{\text{on}})} \text{OFF}$$

$$R_3 : \text{ON} \xrightarrow{k_{\text{syn}}} \text{ON} + \text{mRNA}$$

$$R_4 : \text{mRNA} \xrightarrow{k_{\text{deg}}} \varnothing, \tag{1}$$

where $R_1$ and $R_2$ describe the gene-state switching randomly between the ON and OFF states, and $f_{\text{off}}(t,\boldsymbol{\omega}_{\text{off}})$ and $f_{\text{on}}(t,\boldsymbol{\omega}_{\text{on}})$ are the distribution of dwell time in ON and OFF states, respectively. Specifically, dwell time distribution refers to the probability distribution of the time in which a gene remains in a particular transcriptional state (ON or OFF) before transitioning to the other state. While the gene is in the active state, the synthesis of mRNA molecules is governed by a Poisson process with a constant synthesis rate $k_{\text{syn}}$ ($R_3$). Meanwhile, each mRNA molecule can be stochastically degraded ($R_4$) with a constant degradation rate $k_{\text{deg}}$. Overall, the TXmodel can be determined by a set of parameters, denoted as $\boldsymbol{\theta} = (\boldsymbol{\omega}_{\text{off}}, \boldsymbol{\omega}_{\text{on}}, k_{\text{syn}}, k_{\text{deg}})$.

The TXmodel as an extension of the traditional telegraph model can capture more realistic transcriptional burst kinetics, with BF and BS being two important parameters. The definition of BF is

$$\text{BF} = \frac{1}{\langle \tau_{\text{on}} \rangle + \langle \tau_{\text{off}} \rangle}, \tag{2}$$

where $\langle \tau_{\text{off}} \rangle = \int_0^\infty t f_{\text{off}}(t,\boldsymbol{\omega}_{\text{off}})dt$ and $\langle \tau_{\text{on}} \rangle = \int_0^\infty t f_{\text{on}}(t,\boldsymbol{\omega}_{\text{on}})dt$ are the mean dwell time in OFF and ON states, respectively. BF represents the average time it takes for the promoter to complete one full stochastic cycle between its active and inactive states. This definition is slightly different from the traditional definition $1/(k_{\text{deg}} * \tau_{\text{off}})$, which can be regarded as a simplified version of our definition (*Equation 2*), under the assumptions that $\tau_{\text{on}}$ is negligible and $k_{\text{deg}} = 1$ (i.e., rate parameters are normalized to be dimensionless). Although it is reasonable to neglect activation time $\tau_{\text{on}}$, as it is typically much shorter than inactive time under some conditions, we chose a more complete way to define the BF so that it is applicable to more general situations. Additionally, *Equation 2* has been widely used in recent literature (*Ramsköld et al., 2024*; *Zoller et al., 2018*; *Hoppe et al., 2020*). The definition of BS is

$$\text{BS} = k_{\text{syn}} \langle \tau_{\text{on}} \rangle. \tag{3}$$

BS refers to the average number of mRNA transcripts produced during a single transcriptional activation event of a gene. Notably, the mean transcription level of mRNA $\langle y \rangle$ can be expressed as $\langle y \rangle = k_{\text{syn}} \langle \tau_{\text{on}} \rangle / (\langle \tau_{\text{on}} \rangle + \langle \tau_{\text{off}} \rangle) = \text{BS} \times \text{BF}$.

Without loss of generality, the gamma distribution can be used to characterize non-exponential distributions, that is, $f_{\text{off}}(t,\boldsymbol{\omega}_{\text{off}}) = r_{\text{off}}^{k_{\text{off}}} t^{k_{\text{off}}-1} e^{-r_{\text{off}}t}/\Gamma(k_{\text{off}})$ such that $\boldsymbol{\omega}_{\text{off}} = (r_{\text{off}}, k_{\text{off}})$ and

$f_{\mathrm{on}}(t, \omega_{\mathrm{on}}) = r_{\mathrm{on}}^{k_{\mathrm{on}}} t^{k_{\mathrm{on}}-1} e^{-r_{\mathrm{on}}t} / \Gamma(k_{\mathrm{on}})$ such that $\omega_{\mathrm{on}} = (r_{\mathrm{on}}, k_{\mathrm{on}})$, where $\Gamma(\cdot)$ is Gamma function, $r_{\mathrm{off}}$ and $r_{\mathrm{on}}$ are the rate-parameters of state switching, $k_{\mathrm{off}}$ and $k_{\mathrm{on}}$ are the possible number of reaction steps between OFF and ON state. The rationale for employing the gamma distribution lies in the complex, multi-step nature of promoter state transitions. Biologically, both the 'ON' and 'OFF' states of a promoter encompass multiple underlying sub-states, with transitions governed by a series of molecular events, each characterized by exponentially distributed waiting times. When aggregated, these sequential exponential processes yield non-exponential overall waiting time distributions. The gamma distribution, being a natural analytical extension of such a convolution of multiple exponential distributions, provides a mathematically and biologically appropriate approximation for such processes. Here, in the TXmodel, the state-switching waiting time follows a non-Markovian distribution (i.e., it is not exponentially distributed), indicating that the system exhibits non-Markovian dynamics.

It is a challenge to solve the stationary distribution of this gene expression system directly from a theoretical approach. But we proved that the binomial moments of this system can be obtained in an iterative form, by rationalizing this gene expression system as a queueing theory model. Thus, we can theoretically compute some important summary statistics $s(\theta)$ given a parameter $\theta$, such as mean, variance, kurtosis, and bimodal coefficient (see details in Appendix 1.2 Modeling stochastic gene expression processes).

## Training neural network to solve gene expression model

The stationary distribution of CRN (1) is difficult to be solved analytically. In general, SSAs are used to approximate the solution of chemical master equations (*Boguñá et al., 2014*; *Masuda and Rocha, 2018*). However, the inference process involves updating the parameters, and each update requires a simulation to obtain the stationary distribution, which makes statistical inference time-consuming. We note that deep learning methods may have potential applications in solving this challenge because they are widely used for approximating the solution of partial differential equations, chemical master equations, and non-Markovian chemical master equations (*Jiang et al., 2021*; *Wang et al., 2019*; *Davis et al., 2020*; *Gupta et al., 2021*; *BarSinai et al., 2019*).

Therefore, in this part, we used a neural network to approximately solve CRN (1). This enables CRN (1) to be solved efficiently and accurately. Details will be introduced in the following part.

### Generating synthetic training dataset

To construct the mapping from the parameters of the TXmodel to the stationary distribution, we first need to generate the neural network's training set, that is, a large-scale range of parameter sets as features and corresponding probability distributions as labels. For the features, we used the Sobol algorithm (*Sobol', 1967*) to conduct large-scale sampling to generate the neural network dataset, which can cover the entire parameter space more evenly than random sampling.

For the labels, we perform the SSA of the TXmodel to obtain the stationary distribution given a set of preset parameters. For clarity, the synthetic training dataset is denoted by $\{(\theta_i, P_{\mathrm{simulation},i})\}$, where $\theta_i$ presents the model parameters of TXmodel, and $P_{\mathrm{simulation},i}$ represents the stationary distribution solution obtained by SSA (*Figure 2A*). Specifically, the value range of each parameter component of $\theta_i$ is $k_{\mathrm{off}} \in (1, 15)$, $r_{\mathrm{off}} \in (0.1, 10)$, $k_{\mathrm{on}} \in (1, 15)$, $r_{\mathrm{on}} \in (0.01, 10)$, and $k_{\mathrm{syn}} \in (0.1, 400)$. The experimentally measured values of the mean duration of the OFF state, burst duration, transcriptional burst sizes, and the number of intermediate steps in transcriptional state transitions fall within the specified parameter ranges (*Tunnacliffe and Chubb, 2020*; *Gupta et al., 2022*).

### Deep learning model architecture

We build the basic deep learning model architecture in DeepTX via fully connected neural networks, which has been effectively used to solve approximate solutions from biochemical reaction networks (*Sukys et al., 2022*; *Gupta et al., 2021*). The deep learning model architecture consists of an input layer, hidden layers, and an output layer.

First, the input layer contains five neurons corresponding to the parameters of the mechanism model $\theta_i = (k_{\mathrm{off},i}, r_{\mathrm{off},i}, k_{\mathrm{on},i}, r_{\mathrm{on},i}, k_{\mathrm{syn},i})$. Second, the input layer is followed by $n_{\mathrm{hidden}}$ hidden layers with each layer containing $\boldsymbol{n}_{\mathrm{neurons}} = (n_{\mathrm{neurons},1}, \ldots, n_{\mathrm{neurons},n_{\mathrm{hidden}}})$ neurons. For the neurons in each hidden layer, we implement a nonlinear mapping by the commonly used ReLU activation function (*Figure 2— figure supplement 1*):

$$h_{ij}^{(1)} = \text{ReLU}\left(w_{1j}^{(1)}k_{\text{off},i} + w_{2j}^{(1)}r_{\text{off},i} + w_{3j}^{(1)}k_{\text{on},i} + w_{4j}^{(1)}r_{\text{on},i} + w_{5j}^{(1)}k_{\text{syn},i}\right), \quad (j = 1, 2, \ldots, n_{\text{neurons},1}),$$

$$h_{ij}^{(2)} = \text{ReLU}\left(w_{1j}^{(2)}h_{i1}^{(1)} + w_{2j}^{(2)}h_{i2}^{(1)} + \ldots + w_{n_{\text{neurons},1}j}^{(2)}h_{i,n_{\text{neurons},1}}^{(1)}\right), \quad (j = 1, 2, \ldots, n_{\text{neurons},2}),$$

$$\vdots$$

$$h_{ij}^{(n_{\text{hidden}})} = \text{ReLU}\left(w_{1j}^{(n_{\text{hidden}})}h_{i1}^{(n_{\text{hidden}}-1)} + \ldots + w_{n_{\text{neurons},n_{\text{hidden}}-1}j}^{(n_{\text{hidden}})}h_{i,n_{\text{neurons},n_{\text{hidden}}-1}}^{(n_{\text{hidden}}-1)}\right), \quad (j = 1, 2, \ldots, n_{\text{neurons},n_{\text{hidden}}}),$$

$$(4)$$

where $h_{ij}^{(n_{\text{hidden}})}$ represents the value of the $j$ th neuron of the $n_{\text{hidden}}$ th hidden layer of the $i$ th group of parameters, $w_{kj}^{(n_{\text{hidden}})}$ represents the neural network weight connecting the $k$ th neuron of the $n_{\text{hidden}}$ th hidden layer and the $j$ th neuron of the $(n_{\text{hidden}} - 1)$ th hidden layer.

Third, in the output layer, we aim to allow neurons to approximate synthetic scRNA-seq data generated by TXmodel. Negative binomial distributions are a class of distributions that are widely used in modeling scRNA-seq data and have been shown to account for both the 'drop-out' phenomenon and the fact that the Fano factor is greater than 1 in scRNA-seq data (**Risso et al., 2018**; **Love et al., 2014**). Also, the negative binomial distribution has been employed as an approximate stationary solution to the TXmodel for many gene expression systems (**Sukys et al., 2022**; **Öcal et al., 2022**). We assign weights and parameters $(a_{ij}, m_{ij}, l_{ij})$ to the output layer neurons (**Figure 2—figure supplement 1**):

$$h_{ij}^{(\text{output})} = w_{1j}^{(\text{output})}h_{i1}^{(n_{\text{hidden}})} + w_{2j}^{(\text{output})}h_{i2}^{(n_{\text{hidden}})} + \ldots + w_{n_{\text{neurons},n_{\text{hidden}}}j}^{(\text{output})}h_{i,n_{\text{neurons},n_{\text{hidden}}}}^{(n_{\text{hidden}})}, \quad (j = 1, 2, \ldots, n_{\text{output}}),$$

$$(a_{i1}, a_{i2}, \ldots, a_{in_{\text{NB}}}) = \text{Softmax}\left(h_{i1}^{(\text{output})}, \ldots h_{i,3*j+1}^{(\text{output})}, \ldots, h_{i,3*n_{\text{NB}}-2}^{(\text{output})}\right),$$

$$m_{ij} = \text{Exp}\left(h_{i,3*n_{\text{NB}}-1}^{(\text{output})}\right), \quad j = 1, \ldots, n_{\text{NB}},$$

$$l_{ij} = \text{Sigmoid}\left(h_{i,3*n_{\text{NB}}}^{(\text{output})}\right), \quad j = 1, \ldots, n_{\text{NB}},$$

$$(5)$$

where $h_{ij}^{(\text{output})}$ represents the value of the $j$ th neuron of the output layer of the $i$ th set of parameters, $a_{ij}$ is the weight coefficient of the negative binomial distribution, satisfying $\sum_{j=1}^{n_{\text{NB}}} a_{ij} = 1$, $l_{ij}$, $m_{ij}$ represents the $j$ th parameter of the $j$ th negative binomial distribution of the mixed negative binomial distribution, $n_{\text{NB}}$ is the number of negative binomial distributions. And the mixed negative binomial distributions can be expressed as (**Figure 2—figure supplement 1**):

$$P_{\text{neuralnet},i}(n_{\text{counts}}; \boldsymbol{w}(\boldsymbol{\theta}_i)) = \sum_{j=1}^{n_{\text{NB}}} a_{ij}(\boldsymbol{w}(\boldsymbol{\theta}_i)) \cdot \text{NB}\left(n_{\text{counts}}; m_{ij}(\boldsymbol{w}(\boldsymbol{\theta}_i)), l_{ij}(\boldsymbol{w}(\boldsymbol{\theta}))\right), \quad (6)$$

where $\text{NB}(N = n_{\text{counts}}; l, m) = \binom{n_{\text{counts}}+m-1}{n_{\text{counts}}} l^{n_{\text{counts}}}(1-l)^m$.

## Loss function of DeepTXsolver

Our loss function includes two parts: one part is the KL divergence between the mixed negative binomial distribution generated by neural network $P_{\text{neuralnet},i}$ and label distribution $P_{\text{simulation},i}$ generated by SSA, and the other part is the log difference of the moment statistics between theoretical solution of the TXmodel $s_{ij}$ and the output of the neural network $\hat{s}_{ij}$:

$$\text{Loss} = \sum_{i=1}^{n_{\text{batch}}} \left[ \text{KL}\left(P_{\text{simulation},i}, P_{\text{neuralnet},i}\right) + \lambda \sum_{j=1}^{n_{\text{stats}}} \left(\log\left(\frac{s_{ij}}{\hat{s}_{ij}}\right)\right)^2 \right], \quad (7)$$

where $\text{KL}\left(P_{\text{simulation},i}, P_{\text{neuralnet},i}\right) = \sum_n P_{\text{simulation},i}(n)\log\left(P_{\text{simulation},i}(n)/P_{\text{neuralnet},i}(n)\right)$, and $\lambda$ is the hyperparameter, represents the weight coefficient of the statistical loss, and $\lambda = 0.1$, $n_{\text{batch}}$ represents the number of samples, where each sample is characterized by $(\boldsymbol{\theta}_i, P_{\text{simulation},i})$ in each batch, and $n_{\text{stats}}$ is the number of summary statistics. For the latter part of the loss function, we can still obtain the moment statistic by theoretical derivation, although the analytic distribution is difficult to solve. Adding moment statistics to the loss function can make the neural network more robust (**Figure 2E**). Specifically, we use four statistics here: $s_{i1}$ indicates the mean, $s_{i2}$ denotes the Fano factor, $s_{i3}$ denotes

kurtosis, $s_{i4}$ represents the bimodal coefficient. Specifically, the Fano factor, which normalizes variance by the mean, provides a robust measure of transcriptional noise across genes or conditions with varying expression levels. It's worth mentioning that the statistics here are scalable and we can add more information inside the loss function.

## Training details of DeepTXsolver

Effective model training necessitates the selection of an appropriate optimizer to minimize the loss function. In our case, we employed the Adam optimizer (**Kingma and Ba, 2014**), a widely adopted choice in deep learning due to its robust performance. And each time, a small batch of samples is selected to update the neural network parameters $w$ (**Goodfellow et al., 2016**), and the iteration of all samples is called an epoch. Furthermore, we utilized the Glorot Uniform method to establish an optimal initial point for our neural network parameter (**Glorot and Bengio, 2010**).

Batch size and learning rate are key optimization hyperparameters affecting model convergence. A smaller batch size enhances model generalizability but lengthens training time; hence, a batch size of 64 is chosen for balance (**Keskar et al., 2016**). In the gradient descent process, the learning rate significantly influences the step size, where large rates may induce instability or divergence, and conversely, small rates may precipitate premature convergence to local optima. To counteract this, a decaying learning rate is used, allowing larger early updates for faster convergence and preventing instability in later stages. Further, training of the model stops if it has been trained for over 200 epochs or if the learning rate has been reduced five times. These criteria are adaptive and efficient as further training does not significantly enhance the model's performance.

## Hyperparameter tuning of DeepTXsolver

To obtain a neural network model with accurate prediction and generalization, we compared the model architectures in terms of the number of neurons per layer, the number of hidden layers, and the number of neurons in the output layer, and compared the performance of the model with different dataset sizes. And we applied the Hellinger distance between the true distribution and the predicted distribution by the neural network as an evaluation criterion.

First, we compared the performance of architectures with a single hidden layer with (**Chubb et al., 2006**; **Gaskin et al., 2023**; **Laha et al., 2015**; **Subramanian et al., 2005**) neurons and found that the performance of the model barely increased after increasing to 128 neurons (**Figure 2—figure supplement 2a**). Second, after comparing the performance of architectures with different numbers of hidden layers, we found that a single hidden layer has the best performance (**Figure 2—figure supplement 2b**). Third, comparing the performance of $3i$ ($i = 1, 2, \ldots, 10$) neurons in the output layer, we found that after 12 neurons (4 negative binomial distribution mixtures), the performance of the model was not improved. Particularly, four negative binomial distribution mixtures are able to fit the bimodal nature of the distribution as well as ensure good training efficiency (**Öcal et al., 2022**).

Finally, to obtain a dataset that contributes to convergence as well as model generalization while ensuring training efficiency, we compared the model performance for dataset sizes of (500, 1000, 5000, 10,000, and 15,000) and found that increasing the dataset size further does not significantly improve the model performance beyond 15,000 samples (**Figure 2—figure supplement 2c**). Therefore, we identified a single hidden layer of 128 neurons and an output layer of 12 neurons as the appropriate model architecture, and 15,000 samples as the appropriate training set.

## Inferring burst kinetics from scRNA-seq data with DeepTXinferrer

This section describes how to recover dynamic burst kinetics from static scRNA-seq data. In principle, once we have constructed the mapping from parameters to their corresponding stationary distributions through the pre-trained neural network in the Training neural network to solve gene expression model, we can quickly access the corresponding likelihood functions when updating the parameters during the statistical inference process. However, observed scRNA-seq data is the result of a combination of gene expression processes and sequencing processes, and the current pre-trained neural network can only yield a description of the gene expression process modeled by TXmodel. Therefore, we require extending it into a hierarchical model to depict the more realistic generation mechanism behind the scRNA-seq data (**Sarkar and Stephens, 2021**).

## Modeling observed scRNA-seq data with a hierarchical model

The hierarchical model aims to couple two different processes, specifically, the true gene expression mechanism process and the sequencing process that immediately follows. In terms of a formula, the probability distribution of observations can be represented as a convolution:

$$P(Y = y) = \int_0^\infty P_{\text{measure}}(y \mid x) P_{\text{model}}(x) dx, \tag{8}$$

where $P_{\text{measure}}$ is the measure model and $P_{\text{model}}$ is the expression model.

First, we model the sequencing process. Each cell can be regarded as a pool of mRNA, and $X_{cg}^{(\text{true})}$ represents the true expression amount of the $g$ th gene in the $c$ th cell. $Y_{cg}^{(\text{observed})}$ refers to the observed value obtained by scRNA-seq from the given true expression value $X_{cg}^{(\text{true})}$, satisfying the following conditional probability distribution:

$$Y_{cg}^{(\text{observed})} \mid X_{cg}^{(\text{true})} \sim P_{\text{measure}}(y \mid x), \tag{9}$$

where $P_{\text{measure}}(y \mid x)$ characterizes all noise in the sequencing process, and is generally assumed to be Binomial distribution or Poisson distribution, which has been proven experimentally and theoretically (**Sarkar and Stephens, 2021**; **Wang et al., 2018**). By incorporating an additional sampling probability, denoted as $\alpha_{cg}$, into the sequencing process, we are able to characterize capture efficiency. We make the assumption that intercellular molecules are independent of one another, and that only proportional products are captured and sequenced, following a Binomial distribution:

$$Y_{cg}^{(\text{observed})} \mid X_{cg}^{(\text{true})} \sim P_{\text{measure}}(y \mid x, \alpha_{cg}), \tag{10}$$

where $\alpha_{cg}$ is the sampling probability. Although the experimental capture efficiency ranges from 0.06 to 0.32 (**Zheng et al., 2017**; **Macosko et al., 2015**), we fixed the parameter $\alpha_{cg} = 0.5$ to minimize complexity and unidentifiability issues. Further, we demonstrated that inference across different efficiencies (0.5, 0.3, 0.2) consistently yielded strong correlations between inferred burst size and BF.

Second, we model the true gene expression process, assuming that it satisfies the following probability distribution:

$$X_{cg}^{(\text{true})} \sim P_{\text{model}}(x). \tag{11}$$

The selection of the true gene expression probability distribution is very critical. It needs to meet the following conditions: (1) It can well characterize the transcriptional burst kinetics. (2) The inference derived from this distribution necessitates both efficiency and precision.

In the Modeling stochastic gene expression processes, we have constructed a model of the gene expression process with DNA damage, enabling the model's stationary distribution to encapsulate the bursty dynamics inherent in gene expression. The solution to the model is approximated by a mixed negative binomial distribution, which is obtained from a trained neural network. Importantly, this mixed negative binomial distribution enables efficient inference. Hence,

$$P_{\text{model}}(x) = P_{\text{neuralnet}}(x, \boldsymbol{\theta}). \tag{12}$$

Further, the following hierarchical model can be obtained:

$$Y_{cg}^{(\text{observed})} \mid X_{cg}^{(\text{true})} \sim \text{Binomial}\left(y_{cg} \mid x_{cg}, \alpha_{cg}\right),$$

$$X_{cg}^{(\text{true})} \sim \sum_{j=1}^{n_{\text{NB}}} a_j \text{NB}_j\left(x_{cg}; m_j(\boldsymbol{\theta}), l_j(\boldsymbol{\theta})\right), \tag{13}$$

where $X_{cg}^{(\text{true})}$ represents the random variable of the true expression value, $Y_{cg}^{(\text{observed})}$ represents the random variable of the observed expression value, Binomial represents the binomial distribution, NB represents negative binomial distribution, and $\alpha_{cg}$ represents the sampling probability. Further, we can obtain

$$Y_{cg}^{(\text{observed})} \sim \sum_{x_{cg}=1}^{n_{\text{count}}} \left( \text{Binomial}(y_{cg} \mid x_{cg}, \alpha_{cg}) \sum_{j=1}^{n_{\text{NB}}} a_j \text{NB}_j\left(x_{cg}, m_j(\boldsymbol{\theta}), l_j(\boldsymbol{\theta})\right) \right). \tag{14}$$

It is proved that $Y_{cg}^{(\text{observed})}$ also follows a mixed negative binomial distribution (Appendix 1.3 Training neural network to solve gene expression model), denoted as $P_{\text{obsmodel}}$.

## Estimation and optimization process of DeepTXinferrer

We can use the DeepTX framework to infer scRNA-seq data to obtain the corresponding gene expression burst kinetics through optimization methods. Given the expression counts of gene $g$ in all cells, its corresponding probability distribution $P_{\text{obsdata}}$ can be obtained. We use the Hellinger distance to compare the error between the true probability distribution $P_{\text{obsdata}}$ of genes and the model probability distribution $P_{\text{modeldata}}$. The following optimization problem can be obtained:

$$\hat{\boldsymbol{\theta}} = \arg \min_{\boldsymbol{\theta}} \sum_{m=0}^{n_{\text{count}}} \left( \sqrt{P_{\text{obsmodel}}(y_{cg} = m; \boldsymbol{\theta})} - \sqrt{P_{\text{obsdata}}(y_{cg} = m)} \right)^2, \tag{15}$$

where $n_{\text{count}}$ is the maximum value of mRNA expression and $P_{\text{obsdata}}$ is the density probability for each gene $g$. Specifically, $P_{\text{obsmodel}}$ is a function that is differentiable with respect to parameter $\boldsymbol{\theta}_g$. We can use the gradient descent method to obtain the optimal parameters of the TXmodel and the corresponding distribution of the TXmodel.

## Posterior distribution

The optimization method in the previous section can obtain the parameters of the mechanism model that satisfies the distribution of scRNA-seq data. In this section, we give a method to solve the confidence interval of each optimized parameter (*Gaskin et al., 2023*).

Each iteration of the optimization process can obtain the estimated value $\hat{\boldsymbol{\theta}}^{(k)}$ ($k = 1, \ldots, n_{\text{iteration}}$) and loss value $J(\hat{\boldsymbol{\theta}})$ corresponding to the estimated value. In particular, according to (*Equation 15*), the loss is calculated as follows:

$$J(\hat{\boldsymbol{\theta}}^{(k)}, \boldsymbol{\theta}) = J^{(k)} \left( \hat{P}_{\text{obsmodel}}(\hat{\boldsymbol{\theta}}^{(k)}), P_{\text{obsdata}} \right) = \sum_{m=0}^{n_{\text{count}}} \left( \sqrt{\hat{P}_{\text{obsmodel}}(y = m; \hat{\boldsymbol{\theta}}^{(k)})} - \sqrt{P_{\text{obsdata}}(y = m)} \right)^2. \tag{16}$$

The following posterior probability can be obtained:

$$\pi \left( \hat{\boldsymbol{\theta}}^{(k)} \mid P_{\text{obsdata}} \right) \sim \frac{n^{(k)} \exp \left( -J^{(k)} \left( \hat{P}_{\text{obsmodel}}, P_{\text{obsdata}} \right) \right)}{\sum_{k=1}^{n_{\text{iteration}}} \exp \left( -J^{(k)} \left( \hat{P}_{\text{obsmodel}}, P_{\text{obsdata}} \right) \right)} \pi^{(0)} \left( \hat{\boldsymbol{\theta}}^{(k)} \right), \tag{17}$$

where $\pi^{(0)}$ is the prior, we take it as uniform density, $n^{(k)}$ represents the number of values $\hat{\boldsymbol{\theta}}^{(k)}$ takes during the iteration process, and $\exp \left( -J \left( \hat{P}_{\text{mixture}}, P_{\text{simulation}} \right) \right)$ represents the loss potential of each iteration. Let the posterior probability be $P_{\text{posterior}}(\hat{\boldsymbol{\theta}}^{(k)})$, and the marginal distribution corresponding to each component of parameter $\hat{\boldsymbol{\theta}}^{(k)}$ can be obtained:

$$p \left( \hat{\boldsymbol{\theta}}_j^{(k)} \right) \sim \int P_{\text{posterior}} \left( \hat{\boldsymbol{\theta}}^{(k)} \right) d\hat{\boldsymbol{\theta}}_{-j}^{(k)}. \tag{18}$$

The subscript $-j$ means that we integrate the components of $\hat{\boldsymbol{\theta}}^{(k)}$ except $\hat{\boldsymbol{\theta}}_j^{(k)}$. In particular, to allow the iterative parameters to traverse the entire parameter space, multiple optimizations can be performed with different initial values to improve the performance of the posterior distribution.

## Validation on synthetic data

We conducted an evaluation of the inference performance of the DeepTX framework on synthetic scRNA-seq data, focusing on three key aspects. (2) Model accuracy: This refers to the minimal discrepancy between the distribution corresponding to the inferred parameters and the input distribution. (2) Parameter identifiability: This denotes the proximity of the values of the inferred parameters to the true parameters, suggesting that the model can accurately identify the true parameters. (3) Result robustness: This implies that the outcomes of each inference do not exhibit significant variations.

We created a synthetic dataset $\{(\boldsymbol{\theta}_i^{\text{syn}}, P_i^{\text{syn}})\}(i = 1, 2, \ldots, 500)$ using Sobol sampling and SSA under preset parameters. The synthesized distribution $P_i^{\text{syn}}$ is used as input to the DeepTX framework,

and upon optimization, we obtained estimated parameters and their corresponding distribution $\{(\boldsymbol{\theta}_i^{\text{est}}, P_i^{\text{est}})\}$. We showed that the Hellinger distance between the estimated and synthetic dataset's distribution was minimal, indicating accurate estimations (**Figure 3—figure supplement 1a**). Additionally, the burst kinetics of both the synthetic data and the estimated data can be computed using their respective parameters, $\boldsymbol{\theta}_i^{\text{syn}}$ and $\boldsymbol{\theta}_i^{\text{est}}$. We showed that a strong correlation was observed between the synthetic burst kinetics $\boldsymbol{\gamma}^{\text{syn}} = (\text{BF}^{\text{syn}}, \text{BS}^{\text{syn}})$ and estimated burst kinetics $\boldsymbol{\gamma}^{\text{est}} = (\text{BF}^{\text{est}}, \text{BS}^{\text{est}})$ (**Figure 3—figure supplement 1b, c**), suggesting that burst kinetics are identifiable.

Subsequently, we conducted a robustness verification of the framework's ability to infer burst kinetics. The error was computed as follows:

$$\text{Error}(\boldsymbol{\gamma}^{\text{syn}}, \boldsymbol{\gamma}^{\text{est}}) = \left(\log(\text{BF}^{\text{syn}}) - \log(\text{BF}^{\text{est}})\right)^2 + \left(\log(\text{BS}^{\text{syn}}) - \log(\text{BS}^{\text{est}})\right)^2. \tag{19}$$

Given a set of parameters $\boldsymbol{\theta}_i^{\text{syn}}$, we drew samples of sizes (100, 500, 1000, 1500, and 2000) from the probability distribution $P_i^{\text{syn}}$ corresponding to $\boldsymbol{\theta}_i^{\text{syn}}$. For each sample size, we performed 50 repetitions, and then the DeepTX framework was used to infer the distribution obtained from each sampling, setting different initial values each time. We found that the inferred results are robust, and the robustness increases with the increase in the number of samples (**Figure 3—figure supplement 1d, e**).

## Dataset
### IdU and DMSO dataset
To investigate the effects of IdU on genome-wide gene expression, Desai et al. performed scRNA-seq on mESCs, both with and without IdU treatment. Consequently, they obtained scRNA-seq data for 12481 genes of transcriptomes from 812 cells and 13,780 genes of transcriptomes from 744 cells, respectively. The authors reported that IdU drug treatment increased transcriptional noise genome-wide but did not affect transcriptional mean, further affecting cell fate decisions.

We used the DeepTX framework to infer this set of scRNA-seq data to obtain potential transcriptional burst kinetics and further study the impact of changes in transcriptional burst kinetics on cell fate decisions. To eliminate the impact of technical noise on data inference, we preprocess the data as follows. First, we normalized the data $y_{cg}$ using Seurat, where $y_{cg}$ represents the expression level of the gene $g$ in the cell $c$. The normalization process is that we multiplied each $y_{cg}$ by a scaling factor $S = y_{\text{normalized}}/\sum_{g=1}^{n_{\text{genes}}} y_{cg}$, where $y_{\text{normalized}} = 50{,}000$. Particularly, the scaling factor $S$ is related to the total expression of genes in each cell. Second, after normalizing the expression, we divided it into bins with a unit of 1 to obtain the bin where each expression value is located. Finally, to eliminate the impact of low-expression genes on the inference process, we filtered out genes whose average gene expression is less than 1. After data processing, there are 6082 genes from 812 cells left in our IdU and 6082 genes from 744 cells left in DMSO datasets.

### 5FU dataset
To study the impact of 5FU drug-induced DNA damage on cell fate decisions, Park et al. performed scRNA-seq on colon cancer cells treated with 0, 10, and 50 μM 5FU drug (**Park et al., 2020**) and obtained 8534 genes from 1673 cells, 7077 genes from 632 cells, and 6661 genes from 619 cells, respectively. The authors found that colon cancer cells treated with different doses of 5FU exhibit different transcriptional phenotypes, and these different phenotypes correspond to different cell fate decisions (**Park et al., 2020**).

We used the DeepTX framework to infer these three sets of scRNA-seq data to obtain potential transcriptional burst kinetics, thereby studying the impact of changes in transcriptional burst kinetics caused by different doses of 5FU treatment on cell fate decisions. To eliminate the impact of technical noise on data inference, we preprocess the data in the following steps.

First, we normalize $y_{cg}$ by multiplying a scale factor $S$, where $S = y_{\text{normalized}}/\sum_{g=1}^{n_{\text{genes}}} y_{cg}$, $y_{\text{normalized}} = 2500$. Second, we divide the normalized data into intervals of unit 1 to round the data. Finally, to eliminate the impact of low-expression genes on the inference process, we filtered out genes whose average gene expression is less than 1. After data processing, there are 730 genes from 593 cells left in the control dataset, 174 genes from 593 cells left in our 10 μM 5FU treatment dataset, and 187 genes from 564 cells left in the 50 μM 5FU treatment dataset.

## Acknowledgements

This work was supported by National Key R&D Program of China (Grant No. 2021YFA1302500), Natural Science Foundation of P R China (Grants No. 12171494, No. 12301646 and No. 12501700), Key-Area Research and Development Program of Guangzhou, P R China (Grant No. 2019B110233002), Guangdong Basic and Applied Basic Research Foundation (Grants No. 2022A1515011540, No. 2023A1515110273, and No. 2024A1515012786), and Guangdong Provincial Key Laboratory of Mathematical and Neural Dynamical Systems (Grant No. 2024B1212010004).

## Additional information

### Funding

| Funder | Grant reference number | Author |
| --- | --- | --- |
| National Key Research and Development Program of China | 2021YFA1302500 | Jiajun Zhang |
| Natural Science Foundation of P.R. China | 12171494 | Jiajun Zhang |
| National Natural Science Foundation of China | 12301646 | Zhenquan Zhang |
| National Natural Science Foundation of China | 12501700 | Zihao Wang |
| Key-Area Research and Development Program of Guangzhou, P. R. China | 2019B110233002 | Jiajun Zhang |
| Guangdong Basic and Applied Basic Research Foundation | 2022A1515011540 | Jiajun Zhang |
| Guangdong Basic and Applied Basic Research Foundation | 2023A1515110273 | Zihao Wang |
| Guangdong Basic and Applied Basic Research Foundation | 2024A1515012786 | Zihao Wang |
| Guangdong Provincial Key Laboratory of Mathematical and Neural Dynamical Systems | 2024B1212010004 | Jiajun Zhang |

The funders had no role in study design, data collection, and interpretation, or the decision to submit the work for publication.

### Author contributions

Zhiwei Huang, Songhao Luo, Data curation, Software, Formal analysis, Validation, Investigation, Visualization, Methodology, Writing – original draft, Writing – review and editing; Zihao Wang, Data curation, Validation, Investigation, Visualization, Writing – review and editing; Zhenquan Zhang, Data curation, Formal analysis, Validation, Investigation, Visualization, Writing – review and editing; Benyuan Jiang, Resources, Supervision, Writing – review and editing; Qing Nie, Supervision, Writing – review and editing; Jiajun Zhang, Conceptualization, Resources, Data curation, Formal analysis, Supervision, Funding acquisition, Validation, Investigation, Visualization, Methodology, Writing – original draft, Project administration, Writing – review and editing

### Author ORCIDs

Zhiwei Huang https://orcid.org/0009-0007-1957-3940
Songhao Luo https://orcid.org/0000-0003-1162-9608
Zihao Wang https://orcid.org/0000-0001-5440-843X

Zhenquan Zhang https://orcid.org/0000-0002-2913-4905
Jiajun Zhang https://orcid.org/0000-0001-7107-4814

Joint Public Review: https://doi.org/10.7554/eLife.100623.4.sa1
Author response https://doi.org/10.7554/eLife.100623.4.sa2

## Additional files

### Supplementary files
MDAR checklist

### Data availability
All the scRNA-seq data can be obtained from the public database. The scRNA-seq data of mouse embryonic stem cells (mESC) + DMSO and mESC + IdU data reported here is available at the Gene Expression Omnibus (GEO) database under accession number: GSE176044. The scRNA-seq data of RKO cells with different 5-fluorouracil treatment is available at the Gene Expression Omnibus (GEO) database under accession number: GSE149224. The code for reproducing the presented analysis results is available at the GitHub repository (https://github.com/cellfateTX/DeepTX, copy archived at *cellfateTX and cogitoErgoSum, 2025*).

The following previously published datasets were used:

| Author(s) | Year | Dataset title | Dataset URL | Database and Identifier |
|---|---|---|---|---|
| Desai RV, Weinberger LS | 2021 | A DNA Repair Pathway Can Regulate Transcriptional Noise to Promote Cell Fate Transitions | https://www.ncbi.nlm.nih.gov/geo/query/acc.cgi?acc=GSE176044 | NCBI Gene Expression Omnibus, GSE176044 |
| Park SR, Kang HM, Lee JH | 2020 | Transcriptome landscape of DNA-damage response at single-cell resolution | https://www.ncbi.nlm.nih.gov/geo/query/acc.cgi?acc=GSE149224 | NCBI Gene Expression Omnibus, GSE149224 |

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

## Appendix 1

### 1. DeepTX framework

### 1.1 Modeling gene expression processes in DNA damage

Generally, gene expression switches between ON and OFF states, and the two-state model can model this gene expression process. However, DNA damage will cause multiple different transcription rates during DNA transcription, that is, multiple steps. Therefore, we use a more general model of gene transcription (TX) to model the transcription process, taking DNA damage into account. We first describe the classic telegraph model (CTM), which is a specific case of the TXmodel.

The CTM model is a widely used model for capturing Markovian gene expression burst kinetics. It describes stochastic gene expression as a sequence of four biochemical reactions involving two gene states (ON and OFF), mRNA transcription and degradation:

$$R_1 : \text{OFF} \xrightarrow{k_{\text{off}}} \text{ON}, \quad R_2 : \text{ON} \xrightarrow{k_{\text{on}}} \text{OFF},$$

$$R_3 : \text{ON} \xrightarrow{k_{\text{syn}}} \text{ON} + \text{mRNA}, \quad R_4 : \text{mRNA} \xrightarrow{k_{\text{deg}}} \varnothing, \tag{20}$$

where $k_{\text{off}}$ is the rate at which the gene switches from OFF to ON, $k_{\text{on}}$ is the rate at which the gene switches from ON to OFF, $k_{\text{syn}}$ is the rate of mRNA synthesis and $k_{\text{deg}}$ is the rate of mRNA degradation. In this model, gene switching between active and inactive states is governed by a memoryless Markovian process, where the waiting times for transitions follow exponential distributions.

However, promoter-state switching involves multiple biochemical reaction processes, resulting in the number of effective states of most promoters being greater than two and diverse switching between states. Hence, we consider the more general TXmodel, in which the dwell time of the state switch follows the general distribution. The process of gene transcription is described by following the reaction diagram:

$$R_1 : \text{OFF} \xrightarrow{f_{\text{off}}(t, \omega_{\text{off}})} \text{ON},$$

$$R_2 : \text{ON} \xrightarrow{f_{\text{on}}(t, \omega_{\text{on}})} \text{OFF},$$

$$R_3 : \text{ON} \xrightarrow{k_{\text{syn}}} \text{ON} + \text{mRNA}, \tag{21}$$

$$R_4 : \text{mRNA} \xrightarrow{k_{\text{deg}}} \varnothing,$$

where $R_1$ and $R_2$ describe the gene state switches randomly between the ON and OFF states. The dwell time distributions of OFF and ON states are $f_{\text{off}}(t, \omega_{\text{off}})$ and $f_{\text{on}}(t, \omega_{\text{on}})$, respectively. When the gene is in the active state, $R_3$, the synthesis of mRNA molecules is governed by a Poisson process, with a probability per unit time of a synthesis event equal to $k_{\text{syn}}$. Each mRNA molecule is stochastically degraded, $R_4$, with probabilities per unit time, per molecule, of $k_{\text{deg}}$. Each TXmodel can be determined by a group of parameters, denoted as $\theta = (\omega_{off}, \omega_{on}, k_{\text{syn}}, k_{\text{deg}})$.

### 1.2 Model analysis

In this part, we theoretically solve the statistics of the formula. Let $M(t)$ represent the number of mRNA at time $t$, and $G(t)$ be the state at time $t$. $E_{off}(t), E_{on}(t)$ represents the elapsed time for the gene state to transition to the OFF (ON) state at time $t$. Then, $\{M(t), G(t), E_{off}(t), E_{on}(t); t \geq 0\}$ is a continuous Markov process. Let $p_{off}(n, \tau, t)$ and $p_{on}(n, \tau, t)$ represent $n$ mRNA molecules produced at time $t$ and the elapsed time is the probability density function of $\tau$.

Therefore, we can get

$$p_{\text{off}}(n, \tau, t) \, \Delta \tau = \Pr\{N(t) = n, \ G(t) = \text{OFF}, \ \tau < E_{\text{off}}(t) \leq \tau + \Delta \tau\},$$

$$p_{\text{on}}(n, \tau, t) \, \Delta \tau = \Pr\{N(t) = n, \ G(t) = \text{ON}, \ \tau < E_{\text{on}}(t) \leq \tau + \Delta \tau\}. \tag{22}$$

From the above equation, we can get the following Chapman–Kolmogorov backward equation:

$$p_{\text{off}}(n, \tau + \Delta t, t + \Delta t) = p_{\text{off}}(n, \tau, t)\left(1 - nr_{\text{deg}}\Delta t\right)\left(1 - H_{\text{off}}(\tau)\Delta t\right)$$

$$+ p_{\text{off}}(n+1, \tau, t)(n+1)r_{\text{deg}}\Delta t\left(1 - H_{\text{off}}(\tau)\Delta t\right) + o(\Delta t),$$

$$p_{\text{on}}(n, \tau + \Delta t, t + \Delta t) = p_{\text{on}}(n, \tau, t)\left(1 - nr_{\text{deg}}\Delta t\right)\left(1 - r_{\text{syn}}\Delta t\right)\left(1 - H_{\text{on}}(\tau)\Delta t\right) \tag{23}$$

$$+ p_{\text{on}}(n+1, \tau, t)(n+1)r_{\text{deg}}\Delta t\left(1 - r_{\text{syn}}\Delta t\right)\left(1 - H_{\text{on}}(\tau)\Delta t\right)$$

$$+ p_{\text{on}}(n-1, \tau, t)\left(1 - nr_{\text{deg}}\Delta t\right)r_{\text{syn}}\Delta t\left(1 - H_{\text{on}}(\tau)\Delta t\right) + o(\Delta t),$$

where $H_{\text{off}}(\tau) = f_{\text{off}}(\tau)/S_{\text{off}}(\tau)$ is a hazard rate function with the survival function $S_{\text{off}}(\tau) = \int_{\tau}^{\infty} f_{\text{off}}(t)\,dt$. The definitions of $H_{\text{on}}(\tau)$ and $S_{\text{on}}(\tau)$ are similar.

Next, we focus on the steady-state distributions. Simulations in the main text have verified that the stationary distributions of $p_{\text{off}}(n, \tau, t)$ and $p_{\text{on}}(n, \tau, t)$ exist, and they are denoted by $p_{\text{off}}(n, \tau)$ and $p_{\text{on}}(n, \tau)$, respectively. *Equation 23* converts to the following stationary chemical master equation in the limit of small $\Delta t$ and large $t$,

$$\frac{\partial}{\partial \tau}p_{\text{off}}(n, \tau) = (n+1)k_{\text{deg}}p_{\text{off}}(n+1, \tau) - \left(nk_{\text{deg}} + H_{\text{off}}(\tau)\right)p_{\text{off}}(n, \tau),$$

$$\frac{\partial}{\partial \tau}p_{\text{on}}(n, \tau) = (n+1)k_{\text{deg}}p_{\text{on}}(n+1, \tau) + k_{\text{syn}}p_{\text{on}}(n-1, \tau) - \left(nk_{\text{deg}} + k_{\text{syn}} + H_{\text{on}}(\tau)\right)p_{\text{on}}(n, \tau), \tag{24}$$

with the integral boundary condition

$$p_{\text{off}}(n, 0) = \int_0^{\infty} p_{\text{on}}(n, \tau)H_{\text{on}}(\tau)\,d\tau, \quad p_{\text{on}}(n, 0) = \int_0^{\infty} p_{\text{off}}(n, \tau)H_{\text{off}}(\tau)\,d\tau. \tag{25}$$

Based on *Equation 24* with its boundary conditions, we use the binomial moment method to calculate the mRNA stationary distribution $P(n) = \int_0^{\infty} [p_{\text{off}}(n, \tau) + p_{\text{on}}(n, \tau)d]\tau$ and its statistical characteristics. Binomial moments of the mRNA stationary distribution are defined as $b_m = \sum_{m \geq n}^{\infty} \binom{m}{n}P(n)$, where the symbol $\binom{m}{n}$ represents the combinatorial number. Note that binomial moments converge to zero as their orders go to infinity, and they can be used to reconstruct $|P(n)|$ by $P(n) = \sum_{m=n}^{\infty}(-1)^{m-n}\binom{m}{n}b_m$. After some algebra calculations of *Equation 24*, we can obtain the $n$th binomial moment of mRNA in a recursive form

$$b_n = \frac{1}{n!}\left(\frac{k_{\text{syn}}}{k_{\text{deg}}}\right)^n \sum_{i=0}^{n-1}\binom{n-1}{i}C_i \sum_{j=0}^{n-1-i}\binom{n-1-i}{j}(-1)^{n-1-i-j}\tilde{S}_{\text{on}}\left((n-1-j)k_{\text{deg}}\right), \tag{26}$$

where

$$C_n = \frac{\tilde{f}_{\text{off}}(nk_{\text{deg}})}{1 - \tilde{f}_{\text{off}}(nk_{\text{deg}})\tilde{f}_{\text{on}}(nk_{\text{deg}})}\sum_{i=0}^{n-1}\binom{n}{i}C_i \sum_{j=0}^{n-i}\binom{n-i}{j}(-1)^{n-i-j}\tilde{f}_{\text{on}}\left((n-j)k_{\text{deg}}\right), \tag{27}$$

for $n = 1, 2, \cdots$. Here, $C_0 = \left(\langle \tau_{\text{off}} \rangle + \langle \tau_{\text{on}} \rangle\right)^{-1}$ is equal to the burst frequency. $\tilde{f}(s)$ represents the Laplace transform of function $f(t)$. Especially, $\tilde{S}_{\text{on}}(s) = (1 - \tilde{f}_{\text{on}}(s))/s$ and $\tilde{S}_{\text{on}}(0) = \langle \tau_{\text{on}} \rangle$. According to the relationship between binomial moments and central moments and *Equations 26 and 27*, we obtain the mean and noise of mRNA expression

$$\text{Mean} = \frac{k_{\text{syn}}\langle \tau_{\text{on}} \rangle}{k_{\text{deg}}\left(\langle \tau_{\text{off}} \rangle + \langle \tau_{\text{on}} \rangle\right)},$$

$$\text{CV}^2 = \frac{1}{\text{Mean}} + \frac{\langle \tau_{\text{off}} \rangle}{\langle \tau_{\text{on}} \rangle} + \frac{\langle \tau_{\text{off}} \rangle + \langle \tau_{\text{on}} \rangle}{k_{\text{deg}}\langle \tau_{\text{on}} \rangle^2}\frac{\left(1 - \tilde{f}_{\text{off}}(k_{\text{deg}})\right)\left(1 - \tilde{f}_{\text{on}}(k_{\text{deg}})\right)}{1 - \tilde{f}_{\text{off}}(k_{\text{deg}})\tilde{f}_{\text{on}}(k_{\text{deg}})}. \tag{28}$$

Note that if we let $\tau_{\text{off}}k_{\text{deg}}$ and $\tau_{\text{on}}k_{\text{deg}}$ be the rescaled random variables for OFF- and ON-state dwell times, and $k_{\text{syn}}/k_{\text{deg}}$ be the rescaled mean synthesis rate, the mean expression not only equals the product of the mean synthesis rate and the stationary probability of ON state, but also the product of burst size and burst frequency.

The binomial moment can obtain any order statistics of the model solution. Next, we introduce the six statistics we use in training neural networks: (1) The mean value $\mu_1$ is the most commonly used

indicator in statistics, and it represents the average level of mRNA expression in scRNA-seq data. (2) The noise strength is a measurement of the dispersion of the probability distribution, defined as $\mu_2/\mu_1^2$, where $\mu_2$ is the variance. (3) The Fano factor is another statistic that measures the dispersion of a probability distribution relative to a Poisson distribution, defined as $\mu_2/\mu_1$. (4) The skewness is a description of the symmetry of the distribution, and it is defined as $\mu_3/\mu_2^{3/2}$, where $\mu_3$ is the third central moment. (5) The kurtosis describes whether the peak of the distribution is abrupt or flat, which is defined as $\mu_4/\mu_2^2$, where $\mu_4$ is the fourth central moment. (6) The bimodality coefficient can describe the bimodal distribution, which is usually a critical feature in a dynamical system. Precisely, we can calculate the central moments with the binomial moment:

$$\mu_k(t) = (-b_1(t))^k + \sum_{l=0}^{k-1}\sum_{j=0}^{k-1} R(k,i,j)(j!)\left(b_1(t)\right)^j b_j(t), \tag{29}$$

in which $R(k,i,j) = (-1)^i \binom{k}{i} u(k-i,j)$ with $u(n,k) = \sum_{i=0}^{k}(-1)^{k-i}\binom{k}{i}i^n$ being the Stirling number of the second kind. Therefore, the above summary statistics can be expressed by binomial moments as follows:

$$
\begin{aligned}
\text{Mean} &= b_1, \\[2mm]
\text{NoiseStrength} &= \frac{2b_2 + b_1 - b_1^2}{b_1^2}, \\[2mm]
\text{FanoFactor} &= \frac{2b_2 + b_1 - b_1^2}{b_1}, \\[2mm]
\text{Skewness} &= \frac{6b_3 + 6b_2 + b_1 - 3b_1(2b_2 + b_1) + 2b_1^3}{\left(2b_2 + b_1 - b_1^2\right)^{3/2}}, \\[2mm]
\text{Kurtosis} &= \frac{24b_4 + 36b_3 + 14b_2 + b_1 - 4b_1(6b_3 + 6b_2 + b_1) + 6b_1^2(2b_2 + b_1) - 3b_1^4}{\left(2b_2 + b_1 - b_1^2\right)^2} - 3, \\[2mm]
\text{BimodalityCoefficient} &= \frac{\text{Skewness}^2 + 1}{\text{Kurtosis}}.
\end{aligned}
\tag{30}
$$

It should be noted that we can extend the summary statistics to higher-order moments because our binomial moments can compute arbitrary high-order moment statistics.

## 1.3 Algorithm pipeline of deepTX framework

**Appendix 1—algorithm 1.**

**Dataset:** We use the Sobol algorithm to sample from a preset range of TXmodel parameters to obtain the parameters $\{\boldsymbol{\theta}_i\}$. For each set of $\boldsymbol{\theta}$, the distribution and moments are obtained using the SSA algorithm and binomial moment theory to form the dataset $\left\{(\boldsymbol{\theta}_i, P_{\text{simulation},i}, s_i)\right\}_{i=1,\ldots,N}$.

**Input:** A batch parameter of TXmodel $\{\boldsymbol{\theta}_i\}_{\text{batch}}$.

**Output:** The solution of the TXmodel obtained from the neural network and the corresponding statistics $\left\{(P_{\text{neuralnet},i}, \hat{s}_i)\right\}_{\text{batch}}$.

Initialize the neural network architecture as well as the weights and select hyperparameters.

**For every epoch do**

1. Sampling from the dataset $\left\{(\boldsymbol{\theta}_i, P_{\text{simulation},i}, s_i)\right\}$ to get a batch with parameter set $\left\{(\boldsymbol{\theta}_i, P_{\text{simulation},i}, s_i)\right\}_{\text{batch}}$.
2. Input parameter set $\{\boldsymbol{\theta}_i\}_{\text{batch}}$ to the neural network to get the distribution and statistics predicted by the neural network $\left\{(P_{\text{neuralnet},i}, \hat{s}_i)\right\}_{\text{batch}}$.
3. Train the neural network by the loss function.

$$L = \sum_{i=1}^{n_{\text{batch}}}\left[\text{KL}\left(P_{\text{simulation},i}, P_{\text{neuralnet},i}\right) + \lambda \sum_{j=1}^{n_{\text{stats}}}\left(\log\left(\frac{s_{ij}}{\hat{s}_{ij}}\right)\right)^2\right].$$

**End for**

## 1.4 Derivation of hierarchical negative binomial mixture distribution

Assume the following hierarchical distribution:

$$Y \mid X \sim \text{Binomial}(y \mid x, p),$$

$$X \sim \text{NB}(n, q). \tag{31}$$

Calculate the marginal distribution as follows:

$$
\begin{aligned}
\Pr\{Y = y\} &= \sum_{x=0}^{\infty} P(X = x, Y = y) = \sum_{x=0}^{\infty} P(Y = y \mid X = x) P(X = x) \\
&= \sum_{x=y}^{\infty} \binom{x}{y} \binom{x+n-1}{x} p^y (1-p)^{x-y} (1-q)^n q^x \quad \text{(conditional probability is 0 if } y > x) \\
&= \sum_{x=y}^{\infty} \frac{x!}{y!(x-y)!} \cdot \frac{(x+n-1)!}{x!(n-1)!} \cdot \frac{(n+y-1)!}{(n+y-1)!} p^y (1-p)^{x-y} (1-q)^n q^x \\
&= \sum_{x=y}^{\infty} \binom{x+n-1}{x-y} \binom{n+y-1}{y} p^y (1-p)^{x-y} (1-q)^n q^x q^y q^{-y} \\
&= \binom{n+y-1}{y} (pq)^y (1-q)^n \sum_{x=y}^{\infty} \binom{x+n-1}{x-y} ((1-p)q)^{x-y} \\
&= \binom{n+y-1}{y} \left( \frac{pq}{1-q+pq} \right)^y \left( \frac{1-q}{1-q+pq} \right)^n.
\end{aligned}
\tag{32}
$$

Further,

$$Y \sim \text{NB}\left(n, \frac{pq}{1-q+pq}\right). \tag{33}$$

If we assume that

$$P(X = x) = w_1 P_1(X = x) + w_2 P_2(X = x), \tag{34}$$

where $P_1$ and $P_2$ are both negative binomial distributions, then we can calculate them separately, and the result is the addition of two negative binomial distributions.

## 2. Relationship between noise and burst dynamics

For a TXmodel,

$$
\text{BF} = \frac{1}{\langle \tau_{\text{on}} \rangle + \langle \tau_{\text{off}} \rangle}, \quad \text{BS} = r_{\text{syn}} \langle \tau_{\text{on}} \rangle,
$$

$$
\langle X \rangle = \frac{r_{\text{syn}} \langle \tau_{\text{on}} \rangle}{r_{\text{deg}} (\langle \tau_{\text{on}} \rangle + \langle \tau_{\text{off}} \rangle)} = \frac{\text{BF} \cdot \text{BS}}{r_{\text{deg}}},
$$

$$
\eta_X^2 = \frac{1}{\langle X \rangle} + \frac{r_{\text{deg}} (\langle \tau_{\text{off}} \rangle)^2}{r_{\text{deg}} \langle \tau_{\text{on}} \rangle \langle \tau_{\text{off}} \rangle + \langle \tau_{\text{on}} \rangle + \langle \tau_{\text{off}} \rangle}. \tag{35}
$$

When the mean protein $\langle X \rangle$ is fixed to a constant $M$, BS and BF are inversely proportional. We consider the following cases to indicate that the noise $\eta_X^2$ is a monotonically decreasing (increasing) function about BF (BS).

(1) $\langle \tau_{\text{on}} \rangle$ is fixed.

$\langle \tau_{\text{off}} \rangle$ and $r_{\text{syn}}$ satisfy the following relation,

$$
r_{\text{syn}} = M r_{\text{deg}} \left( 1 + \frac{\langle \tau_{\text{off}} \rangle}{\langle \tau_{\text{on}} \rangle} \right). \tag{36}
$$

On the other hand, BF decreases with an increasing $\langle \tau_{\text{off}} \rangle$, and BS increases with the corresponding increasing $r_{\text{syn}}$. In this case,

$$\eta_{IX}^2 = \frac{1}{M} + \frac{r_{\deg}}{\frac{r_{\deg}\langle\tau_{\rm on}\rangle + 1}{\langle\tau_{\rm off}\rangle} + \frac{\langle\tau_{\rm on}\rangle}{\langle\tau_{\rm off}\rangle^2}} \tag{37}$$

increases with an increasing $\langle\tau_{\rm off}\rangle$. Therefore, the noise $\eta_X^2$ is a monotonically decreasing (increasing) function about BF (BS).

(2) $r_{\rm syn}$ is fixed.

$\langle\tau_{\rm off}\rangle$ and $\langle\tau_{\rm on}\rangle$ satisfy the following relation,

$$\langle\tau_{\rm off}\rangle = K\langle\tau_{\rm on}\rangle, \tag{38}$$

where the constant $K = (1 - C)/C$ and $C = Mr_{\deg}/r_{\rm syn}$. Note that $C \in (0, 1)$, then $K > 0$. Thus, $\langle\tau_{\rm off}\rangle$ is directly proportional to $\langle\tau_{\rm on}\rangle$.

On the other hand, BF decreases with an increasing $\langle\tau_{\rm on}\rangle$ and the corresponding increasing $\langle\tau_{\rm off}\rangle$, and BS increases with an increasing $\langle\tau_{\rm on}\rangle$. In this case,

$$\eta_{IX}^2 = \frac{1}{M} + \frac{r_{\deg}}{\frac{r_{\deg}}{K} + \frac{1}{K\langle\tau_{\rm on}\rangle} + \frac{1}{K^2\langle\tau_{\rm on}\rangle}} \tag{39}$$

increases with an increasing $\langle\tau_{\rm off}\rangle$. Therefore, the noise $\eta_X^2$ is a monotonically decreasing (increasing) function about BF (BS).

(3) The growth of $r_{\rm syn}$ is consistent with the growth of $\langle\tau_{\rm on}\rangle$.

$\langle\tau_{\rm off}\rangle$ can be obtained by

$$\langle\tau_{\rm off}\rangle = K(r_{\rm syn})\langle\tau_{\rm on}\rangle, \tag{40}$$

where $K(r_{\rm syn}) = (r_{\rm syn} - Mr_{\deg})/Mr_{\deg}$ is a monotonically increasing function about $r_{\rm syn}$. Similar to case 2, it is easy to derive that the noise $\eta_X^2$ is a monotonically decreasing (increasing) function about BF (BS).

## 3. The definition of quality score

The quality score (QS) is calculated based on the difference in $z$-scores derived from GSVA (gene set variation analysis) of gene sets upregulated and downregulated during the quiescent phase and is defined as QS = $z$(up genes) − $z$(down genes) (*Wiecek et al., 2023*). $z$(up genes) represents the standardized enrichment score of the gene set upregulated during quiescence in each sample. A higher value indicates that the quiescence-associated upregulated genes are actively expressed, suggesting that the sample is more likely to be in a quiescent (G0) state. $z$(down genes) corresponds to the standardized enrichment score of genes downregulated during quiescence. A lower value implies effective suppression of these genes, which is also consistent with quiescence. The difference score QS serves as an integrated indicator of the quiescent state: A higher value reflects simultaneous activation of quiescence-associated upregulated genes and repression of downregulated genes, indicating a gene expression profile that strongly aligns with the G0/quiescent state. A lower or negative value suggests a deviation from the quiescent signature, potentially reflecting a proliferative state or failure to enter quiescence.

## 4. The rationale for the parameter space

In our study, we considered five parameters: $\theta = (k_{\rm off}, r_{\rm off}, k_{\rm on}, r_{\rm on}, k_{\rm syn})$. The parameters $k_{\rm off}$ and $k_{\rm on}$ represent the number of intermediate reaction steps involved in transcriptional state transitions. These values were sampled uniformly from the range 1–15, which aligns with biological evidence indicating that most genes undergo either direct (single-step) transitions or a small number of intermediate steps, typically fewer than ten (*Tunnacliffe and Chubb, 2020*). This range is sufficient to capture both widely used single-step models and more detailed multi-step mechanisms without introducing biologically implausible complexity.

Among these parameters, $r_{\rm off}$ and $r_{\rm on}$ denote the rate constants governing stochastic transitions between the OFF and ON transcriptional states, respectively. The mean duration of the OFF state, which corresponds to the time between transcriptional bursts, is given by $\langle\tau_{\rm off}\rangle = k_{\rm off}/r_{\rm off}$, and falls

within the range $\langle\tau_{\text{off}}\rangle \in (0.1, 150)$. Experimental measurements report a median value of $\langle\tau_{\text{off}}\rangle$ approximately 3.7 (*Gupta et al., 2022*), which is well contained within this range. Similarly, the mean duration of the ON state, referred to as the burst duration, is defined by $\langle\tau_{\text{on}}\rangle = k_{\text{on}}/r_{\text{on}}$, and spans the interval $\langle\tau_{\text{on}}\rangle \in (0.1, 1500)$. The experimentally observed median value of 0.12 (*Gupta et al., 2022*) confirms that the parameter range adequately captures biologically realistic dynamics.

The parameter $k_{\text{syn}}$ represents the normalized synthesis rate after accounting for molecular degradation. Its range was chosen based on empirical observations of transcriptional burst sizes, which typically vary from single molecules to several dozen (*Gupta et al., 2022*). Considering the relationship $\text{BS} = k_{\text{syn}} \cdot \langle\tau_{\text{on}}\rangle$, the selected range of $k_{\text{syn}}$ ensures that the experimentally observed burst sizes are well represented within the defined parameter space.

