## [Editor Report · eLife Assessment]

This study presents DeepTX, a **valuable** methodological tool that integrates mechanistic stochastic models with single-cell RNA sequencing data to infer transcriptional burst kinetics at genome scale. The approach is broadly applicable and of interest to subfields such as systems biology, bioinformatics, and gene regulation. The evidence supporting the findings is **solid**, with appropriate validation on synthetic data and thoughtful discussion of limitations related to identifiability and model assumptions.

---

## [Referee Report · Joint Public Review]

In this work, the authors present DeepTX, a computational tool for studying transcriptional bursting using single-cell RNA sequencing (scRNA-seq) data and deep learning. The method aims to infer transcriptional burst dynamics-including key model parameters and the associated steady-state distributions-directly from noisy single-cell data. The authors apply DeepTX to datasets from DNA damage experiments, revealing distinct regulatory patterns: IdU treatment in mouse stem cells increases burst size, promoting differentiation, while 5FU alters burst frequency in human cancer cells, driving apoptosis or survival depending on dose. These findings underscore the role of burst regulation in mediating cell fate responses to DNA damage.

The main strength of this study lies in its methodological contribution. DeepTX integrates a non-Markovian mechanistic model with deep learning to approximate steady-state mRNA distributions as mixtures of negative binomial distributions, enabling genome-scale parameter inference with reduced computational cost. The authors provide a clear discussion of the framework's assumptions, including reliance on steady-state data and the inherent unidentifiability of parameter sets, and they outline how the model could be extended to other regulatory processes.

The revised manuscript addresses the original concerns raised by the reviewers, particularly those related to sample size requirements, distributional assumptions, and the biological interpretation of the inferred parameters. The authors have also included an extensive discussion of the limitations of the methodological framework, including the constraints associated with relying on snapshot data, as well as a broader contextualisation of DeepTX within the landscape of existing tools that link mechanistic modelling and single-cell transcriptomics.

Overall, this work represents a valuable contribution to the integration of mechanistic models with high-dimensional single-cell data. It will be of interest to researchers in systems biology, bioinformatics, and computational modelling.

Comments on revisions:

We thank the authors for their thorough revision and for carefully addressing the points raised in the previous review. At this stage, the reviewers have no further concerns.

---

## [Author Response]

The following is the authors’ response to the previous reviews.

**Joint Public Review:**
In this work, the authors present DeepTX, a computational tool for studying transcriptional bursting using single-cell RNA sequencing (scRNA-seq) data and deep learning. The method aims to infer transcriptional burst dynamics-including key model parameters and the associated steady-state distributions-directly from noisy single-cell data. The authors apply DeepTX to datasets from DNA damage experiments, revealing distinct regulatory patterns: IdU treatment in mouse stem cells increases burst size, promoting differentiation, while 5FU alters burst frequency in human cancer cells, driving apoptosis or survival depending on dose. These findings underscore the role of burst regulation in mediating cell fate responses to DNA damage.The main strength of this study lies in its methodological contribution. DeepTX integrates a non-Markovian mechanistic model with deep learning to approximate steady-state mRNA distributions as mixtures of negative binomial distributions, enabling genome-scale parameter inference with reduced computational cost. The authors provide a clear discussion of the framework's assumptions, including reliance on steady-state data and the inherent unidentifiability of parameter sets, and they outline how the model could be extended to other regulatory processes.The revised manuscript addresses many of the original concerns, particularly regarding sample size requirements, distributional assumptions, and the biological interpretation of inferred parameters. However, the framework remains limited by the constraints of snapshot data and cannot yet resolve dynamic heterogeneity or causality. The manuscript would also benefit from a broader contextualisation of DeepTX within the landscape of existing tools linking mechanistic modelling and single-cell transcriptomics. Finally, the interpretation of pathway enrichment analyses still warrants clarification.Overall, this work represents a valuable contribution to the integration of mechanistic models with highdimensional single-cell data. It will be of interest to researchers in systems biology, bioinformatics, and computational modelling.
**Recommendations for the authors:**

We thank the authors for their thorough revision and for addressing many of the points raised during the initial review. The revised manuscript presents an improved and clearer account of the methodology and its implications. However, several aspects would benefit from further clarification and refinement to strengthen the presentation and avoid overstatement.

(1) Contextualization within the existing literatureThe manuscript would benefit from placing DeepTX more clearly in the context of other computational tools developed to connect mechanistic modelling and single-cell RNA sequencing data. This is an active area of research with notable recent contributions, including Sukys and Grima (bioRxiv, 2024), Garrido-Rodriguez et al. (PLOS Comp Biol, 2021), and Maizels (2024). Positioning DeepTX in relation to these and other relevant efforts would help readers appreciate its specific advances and contributions.

We sincerely thank you for this valuable suggestion. We agree that situating DeepTX within the broader landscape of computational approaches linking mechanistic modeling and single-cell RNA sequencing data will clarify its contributions and advances. In this revised version, we have explicitly discussed the comparison and relation of DeepTX in the context of this active area using an individual paragraph in the Discussion section.

Specifically, we mentioned that the DeepTX research paradigm contributes to a growing line of area aiming to link mechanistic models of gene regulation with scRNA-seq data. Maizels provided a comprehensive review of computational strategies for incorporating dynamic mechanisms into single-cell transcriptomics (Maizels RJ, 2024). In this context, RNA velocity is one of the most important examples as it infers short-term transcriptional trends based on splicing kinetics and deterministic ODEs model. However, such approaches are limited by their deterministic assumptions and cannot fully capture the stochastic nature of gene regulation. DeepTX can be viewed as an extension of this framework to stochastic modelling, explicitly addressing transcriptional bursting kinetics under DNA damage. Similarly, DeepCycle, developed by Sukys and Grima (Sukys A & Grima R, 2025), investigates transcriptional burst kinetics during the cell cycle, employing a stochastic age-dependent model and a neural network to infer burst parameters while correcting for measurement noise. By contrast, MIGNON integrates genomic variation data and static transcriptomic measurements into a mechanistic pathway model (HiPathia) to infer pathway-level activity changes, rather than gene-level stochastic transcriptional dynamics (Garrido-Rodriguez M et al., 2021). In this sense, DeepTX and MIGNON are complementary, with DeepTX resolving burst kinetics at the single-gene level and MIGNON emphasizing pathway responses to genomic perturbations, which could inspire future extensions of DeepTX that incorporate sequence-level information.

(2) Interpretation of GO analysisThe interpretation of the GO enrichment results in Figure 4D should be revised. While the text currently associates the enriched terms with signal transduction and cell cycle G2/M phase transition, the most significant terms relate to mitotic cell cycle checkpoint signaling. This distinction should be made clear in the main text, and the conclusions drawn from the GO analysis should be aligned more closely with the statistical results.

We sincerely appreciate you for the insightful comment. We have carefully re-examined the GO enrichment results shown in Figure 4D and agree that the most significantly enriched terms correspond to mitotic cell cycle checkpoint signaling and signal transduction in response to DNA damage, rather than general G2/M phase transition processes. Accordingly, we have revised the main text to highlight the biological significance of mitotic cell cycle checkpoint signaling.

Specifically, we now emphasize two key points: DNA damage and mitotic checkpoint activation are closely interconnected. (1) The mitotic checkpoint serves as a crucial safeguard to ensure accurate chromosome segregation and maintain genomic stability under DNA damage conditions. Activation of the mitotic checkpoint can influence cell fate decisions and differentiation potential (Kim EM & Burke DJ, 2008; Lawrence KS et al., 2015). (2) Sustained activation of the spindle assembly checkpoint (SAC) has been reported to induce mitotic slippage and polyploidization, which in turn may enhance the differentiation potential of embryonic stem cells (Mantel C et al., 2007). These revisions ensure that our interpretation is consistent with the statistical enrichment results and better reflect the underlying biological processes implicated by the data.

(3) Justification for training on simulated dataThe decision to train the model on simulated data should be clearly justified. While the advantage of having access to ground-truth parameters is understood, the manuscript would benefit from a discussion of the limitations of this approach, particularly in terms of generalizability to real datasets. Moreover, it is worth noting that many annotated scRNA-seq datasets are publicly available and could, in principle, be used to complement the training strategy.

We thank you for this insightful comment. We chose to train DeepTXsolver on simulated data because no experimental dataset currently provides genome-wide transcriptional burst kinetics with known ground truth, which is essential for supervised learning. Simulation enables us to (i) generate large, fully annotated datasets spanning the biologically relevant parameter space, (ii) expose the solver to diverse bursting regimes (e.g., low/high burst frequency, small/large burst size, unimodal/bimodal distributions), and (iii) quantitatively benchmark model accuracy, parameter identifiability, and robustness prior to deployment on real scRNA-seq data.

We acknowledge, however, that simulation-based training has inherent limitations in terms of generalizability. Real biological systems may deviate from the idealized bursting model, exhibit more complex noise structures, or display parameter distributions that differ from those in simulations. Moreover, the lack of ground-truth parameters in experimental scRNA-seq datasets prevents an absolute evaluation of inference accuracy. In the future work, publicly available annotated scRNA-seq datasets could be used to complement this simulation-based training strategy and enhance generalizability. We have revised the manuscript to explicitly discuss both the rationale for using simulated data and the potential limitations of this approach.

(4) Benchmarking against external methodsThe performance of DeepTX is primarily compared to a prior method from the same group. To strengthen the methodological claims, it would be preferable to include benchmarking against additional established tools from the broader literature. This would offer a more objective evaluation of the performance gains attributed to DeepTX.

We thank you for this constructive suggestion. We fully agree that benchmarking DeepTX against additional established tools from the broader literatures would provide a more comprehensive and objective evaluation of DeepTX . In the revised manuscript, we have included comparative analyses with other widely used methods, including nnRNA (From Shahrezaei group (Tang W et al., 2023)), txABC (from our group (Luo S et al., 2023)), txBurst (from Sandberg group (Larsson AJM et al., 2019)), txInfer (from Junhao group (Gu J et al., 2025)) (Supplementary Figure S4). The comparative results indicate that our method demonstrates superior performance in both efficiency and accuracy.

(5) Interpretation of Figures 4-6The revised figures are clear and informative; however, the associated interpretations in the main text remain too strong relative to the type of analysis performed. For instance, in Figure 4, it is suggested that changes in burst size are linked to DNA damage-induced signalling cascades that affect cell cycle progression and fate decisions. While this is a plausible hypothesis, GO and GSEA analyses are correlative by nature and not sufficient to support such a mechanistic claim on their own. These analyses should be presented as exploratory, and the strength of the conclusions drawn should be tempered accordingly. Similar caution should be applied to the interpretations of Figures 5 and 6.

We thank you for this important comment. In the revised manuscript, we have carefully moderated the interpretation of the GO and GSEA results in Figures 4, 5, and 6. Specifically, we now present these analyses as exploratory and emphasize their correlative nature, avoiding causal claims that go beyond the scope of the data. The text has been rephrased to highlight the observed associations rather than implying direct causal relationships.

For Figure 4, we emphasize that while it is tempting to hypothesize that enhanced burst size may contribute to DNA damage-related checkpoint activation and thereby influence cell cycle progression and differentiation, our current results only indicate an association between burst size enhancement and pathways involved in DNA damage response and checkpoint signaling.

For Figure 5, we emphasize that although our GO analysis cannot establish causality, the results are consistent with an association between 5-FU-induced changes in burst kinetics and pathways related to oxidative stress and apoptosis. Based on this, we propose a model outlining a potential process through which DNA damage may ultimately lead to cellular apoptosis.

For Figure 6, we emphasize that these enrichment results suggest that high-dose 5FU treatment may be associated with processes such as telomerase activation and mitochondrial function maintenance, both of which have been implicated in cell survival and apoptosis evasion in previous experimental studies. For example, prior work indicates that hTERT translocation can activate telomerase pathways to support telomere maintenance and reduce oxidative stress, which is thought to contribute to apoptosis resistance. While our enrichment analysis cannot establish causality, the observed transcriptional bursting changes are consistent with these reported survival-associated mechanisms.

(6) Discussion section framingThe initial paragraphs of the discussion section make broad biological claims about the role of transcriptional bursting in cellular decision-making. While transcriptional bursting is undoubtedly relevant, the manuscript would benefit from a more cautious framing. It would be more appropriate to foreground the methodological contributions of DeepTX, and to present the biological insights as hypotheses or observations that may guide future experimental investigation, rather than as established conclusions.

We thank you for this insightful comment. We have revised the discussion to clarify and appropriately temper our claims regarding transcriptional bursting. First, we now explicitly recognize that transcriptional bursting is one of multiple contributors to cellular variability, rather than the sole or dominant factor driving cellular decision-making. Second, we have restructured the opening of the discussion to prioritize the methodological contributions of DeepTX, highlighting its strength as a framework for inferring genomewide burst kinetics from scRNA-seq data. Finally, the biological insights derived from our analysis are now presented as correlative observations and potential hypotheses, which may inform and guide future experimental investigations, rather than as definitive mechanistic conclusions.

Small Comments(1) Presentation of discrete distributions: In several figures (e.g., Figure 2B and Supplementary Figures S4, S6, and S8), the comparisons between empirical mRNA distributions and DeepTX-inferred distributions are visually represented using connecting lines, which may give the impression that continuous distributions are being compared to discrete ones. Given the focus on transcriptional bursting, a process inherently tied to discrete stochastic events, this representation could be misleading. The figure captions and visual style should be revised to clarify that all distributions are discrete and to avoid potential confusion. In general, it is recommended to avoid connecting points in discrete distributions with lines, as this can suggest interpolation or comparison with continuous distributions. This applies to Figures 2A and 2B in particular.

We thank you for this valuable suggestion. To prevent any potential misinterpretation of discrete distributions as continuous ones, we have revised the visual representation of the empirical and DeepTXinferred mRNA distributions in Figures 2B, and Supplementary Figures S4, S6, and S8. Specifically, we have replaced the line plots with step plots, which more accurately capture the discrete nature of transcriptional bursting. Additionally, we have updated the figure captions to clearly state that all distributions are discrete.

(2) Transcription is always a multi-step process. While the manuscript aims to model additional complexity introduced by DNA damage, the current phrasing (e.g., on page 5) could be read as implying that transcription becomes multi-step only under damage conditions. This should be clarified.

We thank you for this helpful observation. We agree that transcription is inherently a multi-step process under all conditions. To avoid any possible misunderstanding, we have revised the text to clarify this point.

Specifically, we now explain that many previous studies have employed simplified two-state models to approximate transcriptional dynamics, however, the gene expression process is inherently a multi-step process, which particularly cannot be neglected under conditions of DNA damage. DNA damage can result in slowing or even stopping the RNA pol II movement and cause many macromolecules to be recruited for damage repair. This process will affect the spatially localized behavior of the promoter, causing the dwell time of promoter inactivation and activation that cannot be approximated by a simple two state. Our work adopts a multi-step model because it is more appropriate for capturing the additional complexity introduced by DNA damage.

(3) The first sentence of the discussion section overstates the importance of transcriptional bursting. While it is a key source of variability, it is not the only nor always the dominant one. Furthermore, its role in DNA damage response remains an emerging hypothesis rather than a general principle. The claims in this section should be moderated accordingly.

We thank you for this valuable feedback. In the revised discussion, we have moderated the statements in the opening paragraph to better reflect the current understanding. Specifically, we now acknowledge that transcriptional bursting represents one of multiple sources of variability and is not always the dominant contributor. In addition, we have reframed the role of transcriptional bursting in DNA damage response as an emerging hypothesis, rather than a general principle. To further address this concern, we replaced conclusion-like statements with more cautious, hypothesis-oriented phrasing, presenting our observations as potential directions for future experimental validation.

References

Maizels, R.J. 2024. A dynamical perspective: moving towards mechanism in single-cell transcriptomics. Philos Trans R Soc Lond B Biol Sci 379: 20230049. DOI: https://dx.doi.org/10.1098/rstb.2023.0049, PMID: 38432314

Sukys, A., Grima, R. 2025. Cell-cycle dependence of bursty gene expression: insights from fitting mechanistic models to single-cell RNA-seq data. Nucleic Acids Research 53. DOI: https://dx.doi.org/10.1093/nar/gkaf295, PMID: 40240003

Garrido-Rodriguez, M., Lopez-Lopez, D., Ortuno, F.M., Peña-Chilet, M., Muñoz, E., Calzado, M.A., Dopazo, J. 2021. A versatile workflow to integrate RNA-seq genomic and transcriptomic data into mechanistic models of signaling pathways. PLOS Computational Biology 17: e1008748. DOI: https://dx.doi.org/10.1371/journal.pcbi.1008748, PMID: 33571195

Kim, E.M., Burke, D.J. 2008. DNA damage activates the SAC in an ATM/ATR-dependent manner, independently of the kinetochore. PLOS Genet 4: e1000015. DOI: https://dx.doi.org/10.1371/journal.pgen.1000015, PMID: 18454191

Lawrence, K.S., Chau, T., Engebrecht, J. 2015. DNA damage response and spindle assembly checkpoint function throughout the cell cycle to ensure genomic integrity. PLOS Genet 11: e1005150.DOI: https://dx.doi.org/10.1371/journal.pgen.1005150, PMID: 25898113

Mantel, C., Guo, Y., Lee, M.R., Kim, M.K., Han, M.K., Shibayama, H., Fukuda, S., Yoder, M.C., Pelus, L.M., Kim, K.S., Broxmeyer, H.E. 2007. Checkpoint-apoptosis uncoupling in human and mouse embryonic stem cells: a source of karyotpic instability. Blood 109: 4518-4527. DOI: https://dx.doi.org/10.1182/blood-2006-10-054247, PMID: 17289813

Tang, W., Jørgensen, A.C.S., Marguerat, S., Thomas, P., Shahrezaei, V. 2023. Modelling capture efficiency of single-cell RNA-sequencing data improves inference of transcriptome-wide burst kinetics. Bioinformatics 39. DOI: https://dx.doi.org/10.1093/bioinformatics/btad395, PMID: 37354494

Luo, S., Zhang, Z., Wang, Z., Yang, X., Chen, X., Zhou, T., Zhang, J. 2023. Inferring transcriptional bursting kinetics from single-cell snapshot data using a generalized telegraph model. Royal Society Open Science 10: 221057. DOI: https://dx.doi.org/10.1098/rsos.221057, PMID: 37035293

Larsson, A.J.M., Johnsson, P., Hagemann-Jensen, M., Hartmanis, L., Faridani, O.R., Reinius, B., Segerstolpe, A., Rivera, C.M., Ren, B., Sandberg, R. 2019. Genomic encoding of transcriptional burst kinetics. Nature 565: 251-254. DOI: https://dx.doi.org/10.1038/s41586-018-0836-1, PMID: 30602787

Gu, J., Laszik, N., Miles, C.E., Allard, J., Downing, T.L., Read, E.L. 2025. Scalable inference and identifiability of kinetic parameters for transcriptional bursting from single cell data. Bioinformatics. DOI: https://dx.doi.org/10.1093/bioinformatics/btaf581, PMID: 41131798.